# Interplay between midbrain and dorsal anterior cingulate regions arbitrates lingering reward effects on memory encoding

Kristoffer Carl Aberg [1✉], Emily Elizabeth Kramer[2,3] & Sophie Schwartz[4,5,6]

Rewarding events enhance memory encoding via dopaminergic influences on hippocampal plasticity. Phasic dopamine release depends on immediate reward magnitude, but presumably also on tonic dopamine levels, which may vary as a function of the average accumulation of reward over time. Using model-based fMRI in combination with a novel associative memory task, we show that immediate reward magnitude exerts a monotonically increasing influence on the nucleus accumbens, ventral tegmental area (VTA), and hippocampal activity during encoding, and enhances memory. By contrast, average reward levels modulate feedback-related responses in the VTA and hippocampus in a non-linear (inverted U-shape) fashion, with similar effects on memory performance. Additionally, the dorsal anterior cingulate cortex (dACC) monotonically tracks average reward levels, while VTA-dACC functional connectivity is non-linearly modulated (inverted U-shape) by average reward. We propose that the dACC computes the net behavioral impact of average reward and relays this information to memory circuitry via the VTA.

[1] Department of Neurobiology, Weizmann Institute of Science, Rehovot, Israel. [2] Program in Neurosciences and Mental Health, Hospital for Sick Children, Toronto, ON, Canada. [3] Institute of Medical Sciences, University of Toronto, Toronto, ON, Canada. [4] Department of Neuroscience, University of Geneva, Geneva, Switzerland. [5] Swiss Center for Affective Sciences, University of Geneva, Geneva, Switzerland. [6] Geneva Neuroscience Center, University of Geneva, Geneva, Switzerland. ✉email: kc.aberg@gmail.com

The ability to remember details associated with reward provides an adaptive advantage because it facilitates access to, for example, food and water. Indeed, reward-related information has priority in memory, as evidenced by studies reporting better memory for information presented in trials where a large reward was presented or anticipated (as compared with smaller or no reward)[1–8]. Rewarding events, such as reward feedbacks or cues predictive of reward, initiate phasic dopamine release from the ventral tegmental area (VTA)[9,10], a midbrain region with dopaminergic projections to the hippocampus (HC) whose involvement in episodic memory encoding is well established[11,12]. Thus, the prioritization of reward-related information is presumably mediated by dopaminergic influences on hippocampal plasticity[13,14].

Recent research suggests enhanced memory encoding beyond the rewarded event itself. For example, Mather and Shoeke[15] reported that positive (as compared with negative) feedback enhanced recognition memory for information presented in the immediate trial, as well as in the two subsequent trials. Likewise, we recently demonstrated better memory for associations encoded in trials preceded by many (versus few) rewarded trials[16]. Such modulation of memory for information surrounding contexts associated with frequent rewards likely offers a unique evolutionary advantage. For example, better memory for cues encountered in the vicinity of a location with high food availability increases the probability of successfully finding ones' way to this location in the future. Yet, the neural mechanisms underlying these sustained effects of reward on memory formation remain poorly understood. On the one hand, it has been proposed that average reward is encoded by tonic dopamine levels[17,18], which may result from a sustained mode of dopamine neuron activity that is capable of up- or downregulating phasic dopamine release[19], and would thereby modulate memory encoding. Related to this notion, one elegant study reported that the anticipation of uncertain rewards, a condition which is known to induce a sustained "ramping response" of dopamine neurons[20], increased incidental memory encoding for images presented during the reward anticipation period[21]. On the other hand, recent evidence suggests that the VTA is under top-down control from the anterior cingulate cortex (ACC)[22,23]. The dorsal ACC (dACC) supposedly integrates external and internal motivational factors, such as those related to rewards[24,25]. Moreover, the dACC can modulate reward-related dopamine release by directly interacting with the VTA[26]. Thus, the dACC could contribute to memory encoding by regulating VTA function based on current average reward levels. It is also important to note that the relationship between reward levels and memory might not be linear. In particular, increased levels of dopamine enhance memory encoding only up to a certain point, after which additional dopamine may become detrimental for memory encoding[27,28]. Yet, it is currently unclear whether these effects of dopamine on memory formation relate to changes in VTA function and/or in other brain regions involved in memory formation and motivation, such as the HC and the nucleus accumbens (NAcc).

To elucidate the neural underpinnings of average reward effects on memory encoding, here we combine model-based fMRI with a recently developed associative memory paradigm that allows control of immediate reward magnitude and average reward levels on a trial-wise basis during encoding[16]. To test whether high levels of average reward impair (rather than promote) learning, we include trials that yield higher relative reward as compared with our previous study, in which we found a linear relationship between average reward and memory encoding[16]. We predict that immediate reward magnitude and average reward levels during encoding engage brain regions involved in reward processing and motivation (i.e., the VTA and its downstream target the NAcc) and memory formation (i.e., the HC and the parahippocampal gyrus; PHG), and account for variance in subsequently tested memory performance. However, because the brain loci enabling a non-linear effect of dopamine on episodic memory formation are unclear, we make no specific predictions regarding a differential neuronal representation of linear versus nonlinear effects of immediate and average reward.

In brief, we show that immediate reward-feedback magnitude has a monotonically increasing influence on NAcc, VTA, and hippocampal activity during encoding, and enhances memory retention. Critically, average reward levels during encoding exert a nonlinear modulation (i.e., an inverted U-shape) on: (1) activity in the VTA, the HC, and the PHG, (2) functional connectivity between the dACC and the VTA, and (3) subsequently tested memory performance. By contrast, dACC activity negatively correlates with average reward levels. While the present study confirms a sustained influence of reward feedback on memory encoding, via its modulation of average reward levels and phasic responsivity of dopamine circuitry, these results also support the notion that dACC–VTA interactions regulate motivated behavior.

## Results

**Brief task description.** During fMRI scanning, 34 participants performed an associative memory task consisting of one memory encoding session followed by a memory test session. In each trial of the encoding session, participants were presented with the face of a cartoon character and one pair of objects (Fig. 1a), and guessed which of the two objects the character preferred. Subsequent feedback indicated how much the character liked the selected object: very much (+5 feedback), moderately (+1 feedback), or disliked (−1 feedback; Fig. 1b, c). Participants were instructed to learn the preferences that six different characters had for ten different object pairs, because the memory for the preferences would later be tested. Unbeknownst to participants, the feedbacks were predetermined, such that two characters each received on average high, medium, or low levels of reward (as determined by the ratio between the number of +5 and +1 feedbacks, see Fig. 1d). In order to fully control the average level of reward, each character–object pair was shown only once and all ten preferences for one character was encoded before the ten preferences for the next character was encoded, and so forth (i.e., 60 encoding trials in total). The memory test session was largely similar to the encoding session, but no feedback was provided and the order of character–object pairs was pseudorandomized. Points corresponded to a monetary bonus provided at the end of the experiment (see "Methods" for further details).

**Memory performance.** Memory performance was assessed in the test session, administered 20 min following the end of the encoding session, during which participants had to recall and select the preferred object in each of the ten object pairs for each of the six characters. Performance was calculated as the proportion of correct selections of character–object associations encoded during positive (+1, +5) feedback and correct rejections of associations encoded during negative (−1) feedback. Memory performance was analyzed for the different types of reward presented during the encoding, i.e., three levels of Feedback value (−1, +1, +5) and three levels of Average reward (Lo, Me, Hi). A linear mixed effects model with fixed effects Feedback value (−1, +1, +5) and Average reward (Lo, Me, Hi) and participant as random effect revealed a significant main effect of Feedback value [Fig. 1e; $F(2, 288) = 17.905$, $p < 0.001$, ANOVA] because memory was better for character–object associations encoded during +5 feedback [mean ± SEM: $0.739 ± 0.028$] as compared with +1

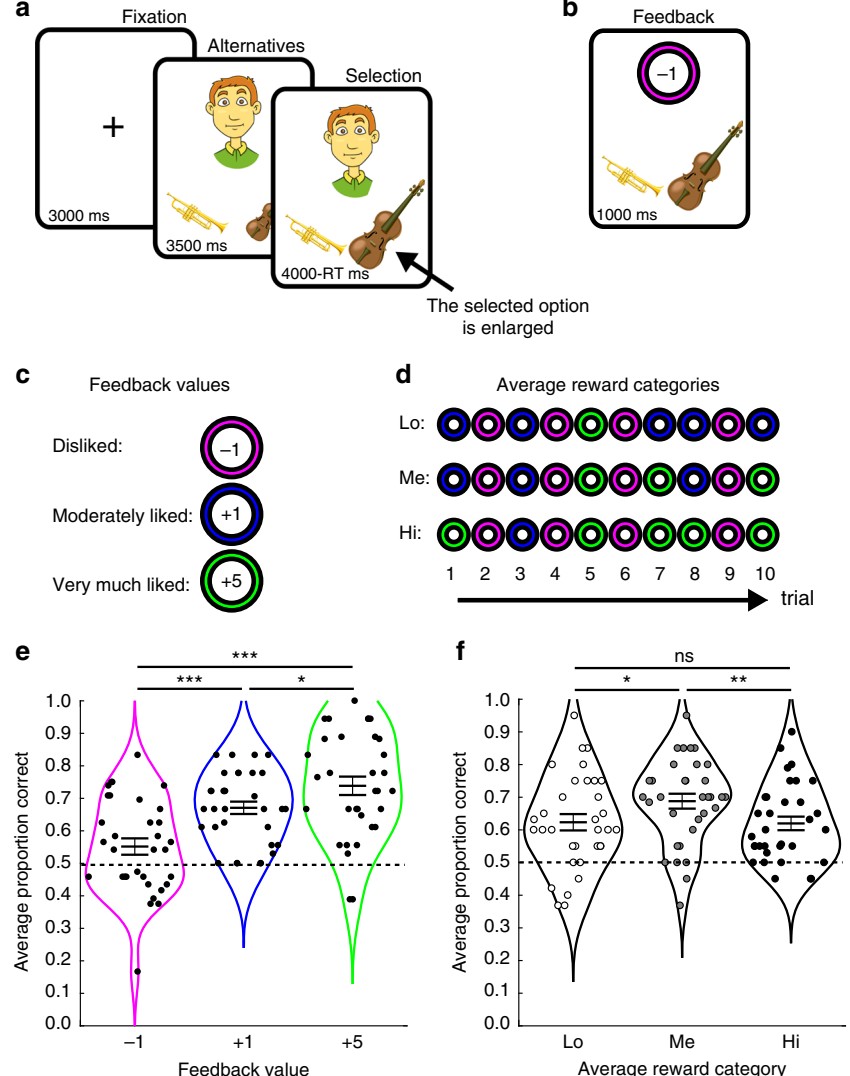

**Fig. 1 Stimuli and behavioral results. a** During both encoding and testing, each trial started with the presentation of a cartoon character together with two objects. During encoding trials, participants randomly selected one object and then received a feedback indicating how much the selected object was liked by the character (see **b**). **c** During encoding, three different feedback values were presented indicating whether a selected object was disliked (magenta circle, −1), moderately liked (blue circle, +1), or very much liked (green circle, +5). **d** Average reward was the average of ten feedback values displayed for a given character. Two characters were associated with high (Hi) average reward (one +1, five +5s), two with medium (Me) average reward (three +1s, three +5s), and two with low (Lo) average reward (five +1s and one +5). **e** Memory performance (proportion correct) increased monotonically as a function of feedback value. **f**. Memory performance increased nonlinearly (inverted U-shape) as a function of average reward. \*\*\*$p < 0.001$, \*\*$p < 0.01$, \*$p < 0.05$, ns not significant ($p > 0.05$), indicates the $p$ value for paired $t$-tests. Horizontal lines in **f** and **g** indicate mean ± SEM. Source data are provided as a Source Data file. The images used to illustrate the paradigm in **a**, **b** were obtained from https://publicdomainvectors.org/ and http://www.freestockphotos.biz/.

feedback [mean ± SEM: 0.671 ± 0.019, $t(32) = 2.545$, $p = 0.016$, $d = 0.490$, 95% CI = 0.014–0.122, paired $t$-test] and −1 feedback [mean ± SEM: 0.552 ± 0.025, $t(32) = 5.728$, $p < 0.001$, $d = 1.214$, 95% CI = 0.120–0.253, paired $t$-test]. Associations encoded during +1 feedback were also better remembered as compared with −1 feedback [$t(32) = 4.039$, $p < 0.001$, $d = 0.929$, 95% CI = 0.059–0.179, paired $t$-test]. When tested separately, memory performance was above change (memory performance > 0.5) for all three types of feedback value [all three $p$ values < 0.05, one-sample $t$-test].

The main effect of Average reward was also significant [Fig. 1f; $F(2,288) = 4.792$, $p = 0.009$ ANOVA], with the highest memory performance for associations encoded during Me average reward [mean ± SEM: 0.688 ± 0.023], both as compared with Lo average

reward [mean ± SEM: 0.623 ± 0.025, $t(32) = 2.240$, $p = 0.032$, $d = 0.470$, 95% CI = 0.006–0.123, paired $t$-test], and Hi average reward [mean ± SEM: 0.620 ± 0.021, $t(32) = 3.230$, $p = 0.003$, $d = 0.544$, 95% CI = 0.025–0.111, paired $t$-test]. There was no difference in memory performance between Lo and Hi average reward [$t(32) = −0.138$, $p = 0.891$, $d = 0.027$, 95% CI = −0.049–0.056, paired $t$-test]. The interaction between Feedback value and Average reward was not significant [$F(4,288) = 1.008$, $p = 0.404$, ANOVA], suggesting independent contributions of Feedback value and Average reward to memory formation. The main effects of Feedback value and Average reward remained significant when controlling for presentation order during encoding, presentation order during testing, and cartoon character identity (see Supplementary Note 1).

**Computational modeling**. Computational modeling was used to provide a more fine-grained (i.e., trial-wise) description of the relationship between the different types of reward and memory formation. The fits of different computational models to behavior are displayed in Table 1 (for full details of these models, see "Methods"). The best fit to behavior was provided by the $R_{fb+\bar{r}^2}$ model, which assumes linear and quadratic influences from feedback value and average reward on memory formation, respectively [the average Akaike's information criterion (AIC) of this model was significantly smaller as compared with the average AIC of each of the other models, all $p$ values < 0.05, paired $t$-tests]. The $R_{fb+\bar{r}^2}$ model also provided the most parsimonious fit as compared with models fitting initial average reward levels and learning rates for each character separately (see Supplementary Note 2).

Figure 2a illustrates how average reward levels ($\bar{r}$) change as a function of recently received feedback. As expected, the highest and the smallest accumulated level of average reward were obtained in the Hi and Lo average reward categories, respectively (Fig. 2b). Figure 2c illustrates the contribution of average reward levels to the encoding probability ($p_E$). $p_E$ increased with increasing levels of average reward, until the "optimal" level of average reward ($w$), at which point additional levels of average reward reduced the encoding probability (Fig. 2d).

The fit of the $R_{fb+\bar{r}^2}$ model to behavioral data is displayed in Fig. 2e, f, and shows the same significant effects as actual behavior (Fig. 1f, g; see Supplementary Note 3). Of note, the level of average reward was not constrained to particular character identities, as the $R_{fb+\bar{r}^2}$ model provided a better fit as compared with other models fitting (1) one baseline value of average reward which was reset whenever a new character was presented during encoding (see the $R_{fb+\bar{r}^2}$ with $C_{\bar{r}0}$ model), or (2) six different baseline levels of average reward (i.e., one for each character; see Supplementary Note 2). Moreover, as would be expected, the $R_{fb+\bar{r}^2}$ model predicted a higher encoding probability for trials subsequently classified as hits [mean ± SEM: 0.679 ± 0.018], as compared with misses [mean ± SEM: 0.574 ± 0.015, $t(32) = 9.620$, $p < 0.001$, $d = 1.136$, 95% CI = 0.083–0.128, paired $t$-test]. Thus, the computational modeling approach also confirmed the hypothesized link between average reward and memory formation.

**fMRI**. To determine the neuronal correlates of the different reward types during memory encoding, we combined fMRI data with trial-by-trial estimates of feedback values and average reward levels obtained from the $R_{fb+\bar{r}^2}$ model. Because feedback values and average reward levels respectively impacted subsequent memory performance in a monotonically increasing and in a nonlinear fashion, we predicted that blood-oxygen-level-dependent (BOLD) signal in reward- and memory-related brain regions should be similarly modulated by feedback value (i.e., monotonically increasing) and average reward (i.e., in an inverted U-shape fashion).

**Positive correlations between BOLD and feedback values**. An increase in feedback value was reflected in increasing BOLD signal in the bilateral NAcc and in bilateral clusters extending into both the HC, the PHG, and the amygdala (Fig. 3 and Table 2). However, given the well-known role of the VTA in reward processing and reward-related memory enhancements, it was surprising that no voxels with the VTA ROI tracked feedback value. To test whether this null-result may have been related to the selection of an anatomically defined VTA ROI, in combination with a strict requirement for the number of overlapping voxels across participants (50%; see "Methods"), we performed an

**Table 1 Model fits.**

| Model | LLE | AIC | $C_o$ | $C_{fb}$ | $C_{\bar{r}}$ | $C_{fb\bar{r}}$ | $C_{\bar{r}^2}$ | $v$ | $w$ | $C_{\bar{r}0}$ |
|---|---|---|---|---|---|---|---|---|---|---|
| $R_{null}$ | 55.82 ± 0.93 | 113.65 ± 1.86 | 0.59 ± 0.07 | — | — | — | — | — | — | — |
| $R_{fb}$ | 53.64 ± 1.11 | 111.28 ± 2.22 | 0.38 ± 0.07 | 0.65 ± 0.09 | — | — | — | — | — | — |
| $R_{\bar{r}}$ | 54.99 ± 0.89 | 115.98 ± 1.78 | 0.55 ± 0.09 | — | 186.91 ± 117.71 | — | — | 0.46 ± 0.08 | — | — |
| $R_{fb+\bar{r}}$ | 52.78 ± 1.06 | 113.56 ± 2.11 | 0.24 ± 0.10 | 0.70 ± 0.09 | 113.23 ± 98.40 | — | — | 0.47 ± 0.08 | — | — |
| $R_{fb+\bar{r}+fb\times\bar{r}}$ | 52.12 ± 1.09 | 114.23 ± 2.18 | 0.23 ± 0.10 | 0.71 ± 0.08 | 32.37 ± 29.59 | 0.60 ± 0.07 | — | 0.21 ± 0.15 | — | — |
| $R_{fb+\bar{r}^2}$ [a] | 48.84 ± 1.46 | 107.67 ± 2.93 | 0.44 ± 0.09 | 0.61 ± 0.10 | — | — | −42.73 ± 12.47 | 0.44 ± 0.07 | 0.43 ± 0.09 | — |
| $R_{fb+\bar{r}^2}$ with $C_{\bar{r}0}$ | 49.56 ± 1.35 | 111.12 ± 2.70 | 0.44 ± 0.09 | 0.59 ± 0.10 | — | — | −320.10 ± 147.7 | 0.42 ± 0.07 | 0.45 ± 0.09 | 0.46 ± 0.07 |

$C_o, C_{fb}, C_{\bar{r}}, C_{\bar{r}0}, C_{fb\bar{r}}, C_{\bar{r}^2}, v,$ and $w$ are parameters fitted to each individual's behavior.
LLE, negative log-likelihood estimate, AIC, Akaike's information criterion.
[a]The model providing the most parsimonious fit to behavior, as indicated by significantly lower AIC values. Mean ± SEM.

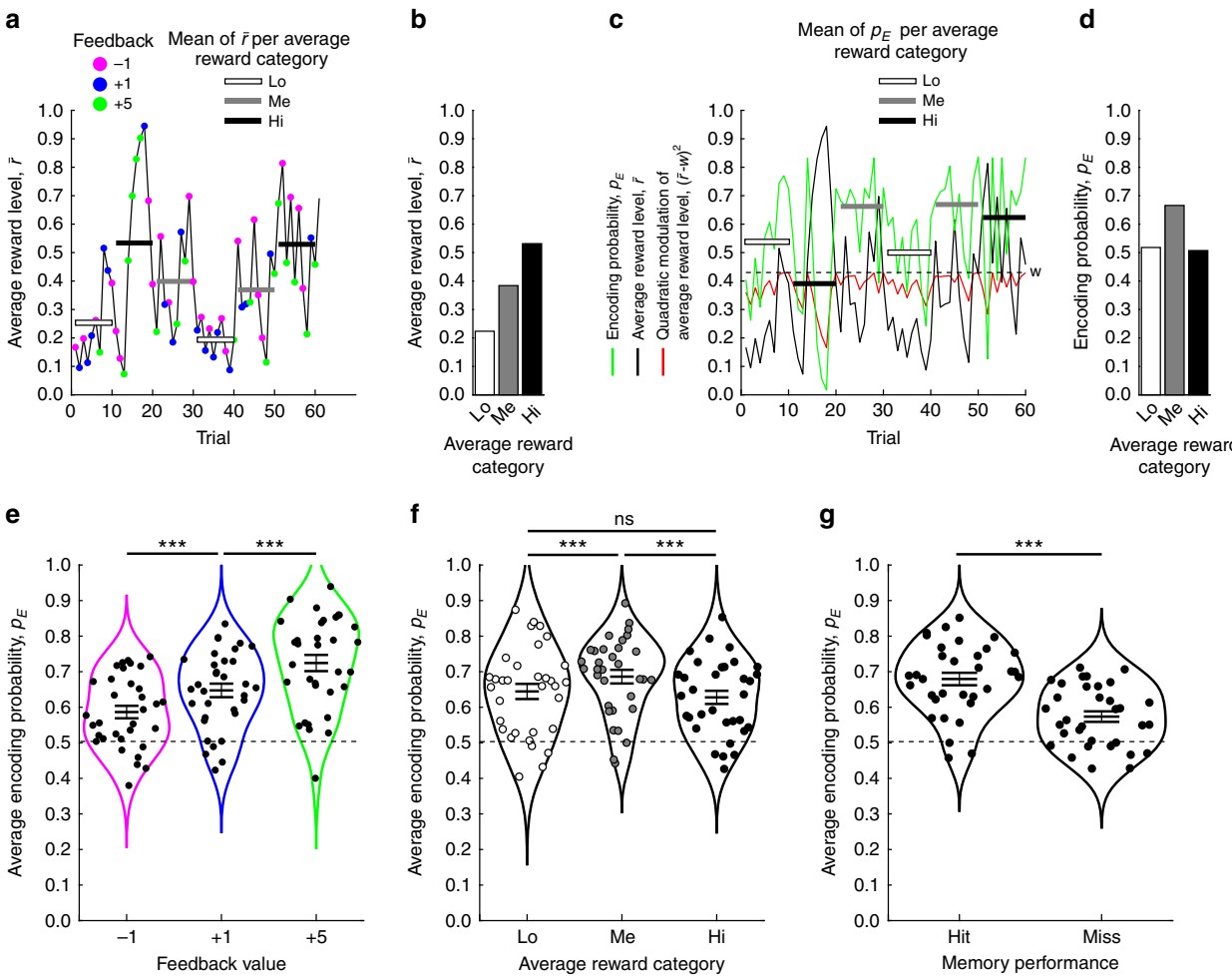

**Fig. 2 Computational model. a–d** illustrates how parameter values in the computational model derive from the behavioral data, here for one representative participant. **a** The average reward level ($\bar{r}$, black line) tracks preceding feedback values as an exponential running average. The average $\bar{r}$ of ten trials within the different average reward categories (Hi, Me, and Lo) are respectively indicated by white, gray, and black bars. **b**. The average $\bar{r}$ across average reward categories in (**a**). **c** When the encoding probability ($p_E$, green line) is nonlinearly (inverted U-shape) modulated by average reward levels ($\bar{r}$, black line), $p_E$ increases when $\bar{r}$ approaches the optimal level of average reward ($w$, dashed line). By contrast, $p_E$ decreases whenever $\bar{r}$ becomes larger or smaller than $w$. To further illustrate the nonlinear contribution of $\bar{r}$ to $p_E$, the quadratic term $(\bar{r} - w)^2$ is displayed as a red line. White, gray, and black bars respectively illustrate the average $p_E$ of the ten trials within each average reward category. **d** The average $p_E$ across the average reward categories in (**c**). **e** Across participants, the most parsimonious model indicates that the encoding probability $p_E$ increases monotonically as a function of feedback value, but is nonlinearly modulated by average reward (**f**). **g** On average the model predicts higher encoding probabilities for subsequent hits as compared with misses. ***$p < 0.001$, ns not significant (i.e., $p > 0.05$), indicates the $p$ value for paired $t$-test. The horizontal lines in **e**, **f**, and **g** indicate mean ± SEM. Source data are provided as a Source Data file.

additional analysis using a conservative ROI approach in a functionally defined VTA ROI. This VTA ROI was based on coordinates obtained from a previous study looking at reward-related memory enhancements[3], and we previously used this VTA ROI to test prediction error encoding in the dopaminergic midbrain[29]. This supplementary analysis revealed that BOLD signal in this VTA ROI indeed correlated with feedback values (see Supplementary Note 4).

Because feedback values modulated memory performance (Figs. 1f and 2e), these results are in accordance with previous studies showing that reward-related effects on memory encoding are associated with increased activity in the NAcc, the VTA, the HC, and the PHG[3,5]. Sagittal views of these activations are provided in the Supplementary Note 6.

**Non linear modulation of BOLD by average reward levels.** BOLD signal was modulated in an inverted U-shape fashion by average reward in the VTA, the amygdala, and in bilateral HC/

PHG clusters (Fig. 4 and Table 3). Because average reward levels contributed significantly to memory encoding (inverted U-shape; Figs. 1g and 2f), this result suggests that average reward influences memory encoding by engaging dopaminergic memory circuitry. Sagittal views of these activations are provided in the Supplementary Note 6. Additional supplementary analyses, using a more conservative ROI approach, confirmed the robustness of the effect of average reward on VTA activity across different VTA ROIs and smoothing kernels (see Supplementary Note 5). Of note, a separate fMRI analysis confirmed that average reward had no impact on BOLD signal during stimulus presentation (see Supplementary Note 7). Moreover, the conjunction between the BOLD signal modulated by average reward (inverted U-shape) and feedback value revealed that no voxels in the a priori ROIs were significantly modulated by both feedback value and average reward (the conjunction was tested versus the "conjunction null hypothesis"[30] with an initial search threshold of $p = 0.001$).

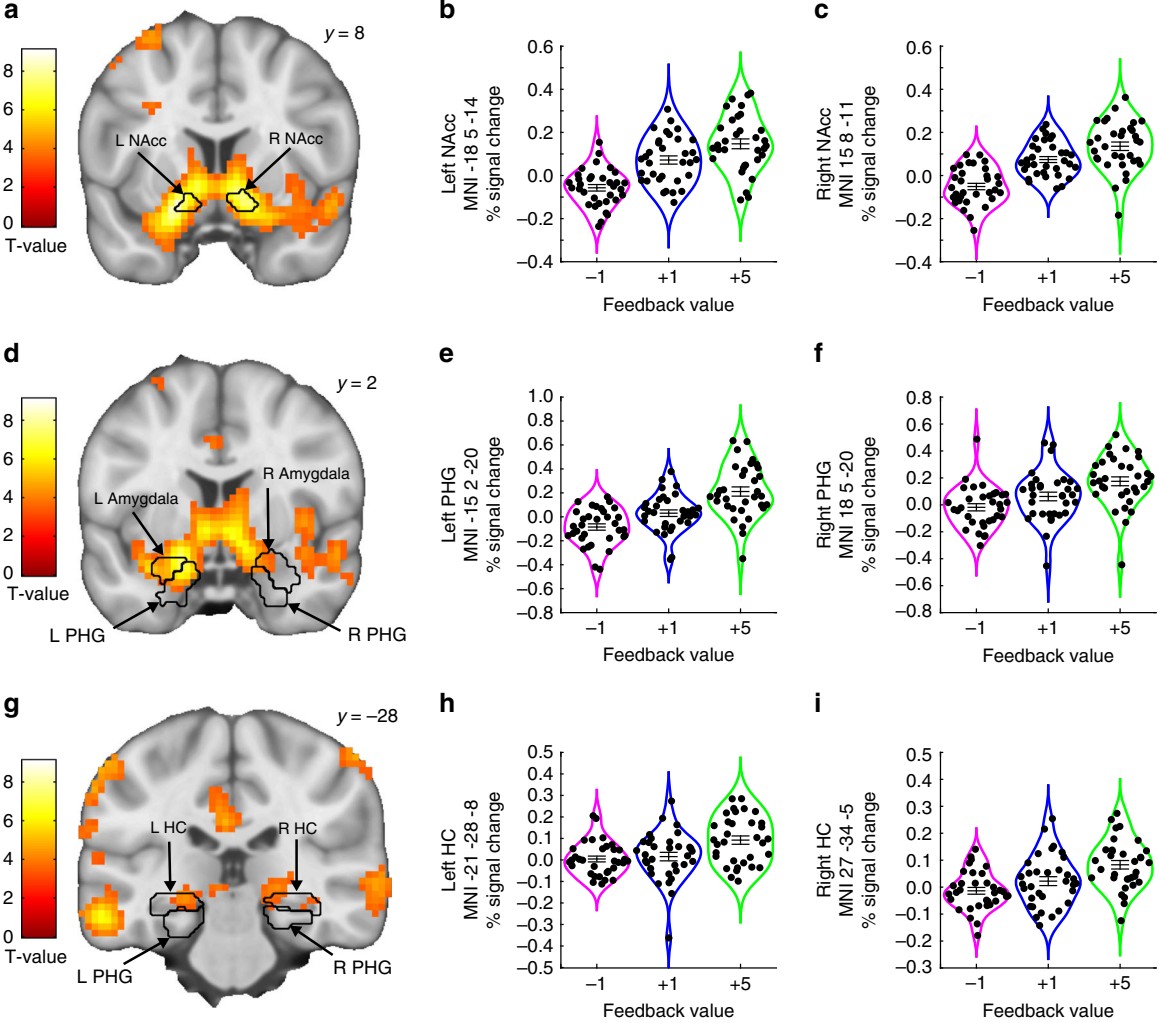

**Fig. 3 Linear BOLD signal increases as a function of feedback value.** BOLD signal in the bilateral NAcc ROI (**a**–**c**), and the HC/PHG ROIs (**d**–**i**) increased monotonically with feedback value. For display purposes, the violin plots show the average % signal change extracted from 3-mm spheres centered on peak voxel coordinates for each significant activation cluster. The horizontal lines indicate mean ± SEM. The BOLD signal is shown using an uncorrected threshold of $p < 0.001$. NAcc nucleus accumbens ROI, HC hippocampus ROI, PHG parahippocampal gyrus ROI, Amygdala amygdala ROI. Source data are provided as a Source Data file. T statistics were obtained from $t$-tests.

**VTA–dACC interaction is modulated by average reward levels**. The dACC integrates different motivational factors, such as reward and effort[24,25], and determines motivated behavior via top-down modulations on the VTA[22,23]. It has recently been proposed that the dACC both keeps track of environmental states, such as current availability of reward or effort needed to obtain a goal, and regulates neuromodulatory systems (including dopaminergic reward signals in the VTA) in order to respond appropriately to changes in the environment[26]. In the context of the present study, we predicted that the dACC would encode average reward levels (i.e., environmental tracking) and communicate the behavioral impact of average reward to the VTA (i.e., neuromodulatory regulation). Specifically, we investigated these hypotheses by testing for linear correlations between average reward and dACC BOLD signal and nonlinear modulations of VTA–dACC functional connectivity by average reward.

Using a generalized psychophysiological interaction (gPPI) approach (seed ROI defined as the voxels within the VTA ROI that correlated non linearly with average reward; Fig. 5a, b), we confirmed that the functional connectivity between the VTA and the dACC was modulated in a nonlinear fashion by average reward (inverted U-shape; Fig. 5a, c; peak MNI $x y z = 3\ 35\ 13$;

$T(32) = 3.751$; $pSVC = 0.037$). Moreover, dACC BOLD signal significantly and linearly correlated with average reward levels (Fig. 5d, e; peak MNI $x y z = 0\ 32\ 22$; $T(32) = 4.155$; $pSVC = 0.016$).

## Discussion

We investigated the influence of reward feedback and average accumulated reward on associative memory formation. Three different levels of feedback value (−1, +1, +5) and average reward (Lo, Me, Hi) during encoding were tested. Model-based fMRI was used to assess how memory performance related to trial-by-trial changes in neural activity under these reward regimes during encoding.

Memory performance increased monotonically for character–object associations encoded during −1, +1, and +5 feedbacks, respectively. This effect was mirrored by activity in the NAcc, the VTA, the HC, and the PHG at encoding. These memory effects primarily depended on the feedback value at encoding (and not the expected reward at memory recall) because associations encoded during +1 feedback were better remembered than those encoded during −1 feedback, although correctly

**Table 2 BOLD signal showing positive correlation with increasing feedback values in a priori ROIs and the amygdala ROI.**

| | Hemisphere | MNI peak coordinate | | | T(32) | $P_{FWE, SVC}$ |
|---|---|---|---|---|---|---|
| | | x | y | z | | |
| Positive correlation with feedback value | | | | | | |
| Reward mask | | | | | | |
| Nucleus accumbens | Left | −18 | 5 | −14 | 9.085 | <0.001 |
| Nucleus accumbens | Right | 15 | 8 | −11 | 7.593 | <0.001 |
| VTA[a] | | | | | | |
| Memory mask | | | | | | |
| Hippocampus/parahippocampal gyrus | Left | −15 | 2 | −20 | 6.724 | <0.001 |
| | | −24 | −13 | −14 | 4.310 | 0.031 |
| | | −21 | −7 | −20 | | 0.032 |
| Hippocampus/parahippocampal gyrus | Left | −21 | −28 | −8 | 4.329 | 0.030 |
| Hippocampus/parahippocampal gyrus | Right | 27 | −34 | −5 | 5.004 | 0.006 |
| | | 18 | 5 | −20 | 4.522 | 0.019 |
| Amygdala ROI (post-hoc) | | | | | | |
| Amygdala[b] | Left | −18 | 2 | −23 | 6.147 | <0.001 |
| Amygdala[b] | Right | 18 | 2 | 20 | 4.069 | 0.017 ns |

$p_{FWE, SVC}$ indicates the p value resulting from family-wise error (FWE) small volume correction (SVC) on peak voxel activity within a priori ROIs and the amygdala ROI. T-statistics were obtained from t-tests.
ns, not significant.
[a]No voxels within the anatomically defined VTA ROI included in the "reward mask" were significantly activated by feedback value. However, a complimentary ROI analysis using a slightly different and functionally defined VTA ROI showed that VTA BOLD signal significantly tracked feedback value (see Supplementary Note 4 for details).
[b]The amygdala was not part of the initial hypotheses, thus a stricter (Bonferroni-corrected) statistical threshold was applied in order to infer any involvement of the amygdala ($\alpha = 0.0167$).

remembering both types of trials in the memory test would yield +1 point. One plausible explanation for these results can be derived from studies showing that unexpected positive or negative feedbacks elicit phasic increases (bursts) or decreases (dips) in dopamine neuron activity, respectively[9,10]. Dopamine neurons project to the NAcc, which is anatomically and functionally linked to the HC[3,5,31,32]. Accordingly, differences in phasic dopamine neuron activity during encoding may explain why associations are better remembered following +1 feedback, as compared with −1 feedback (and similarly why +5 feedbacks caused even better memory performance)[16]. The monotonically increasing relationship between three levels of feedback value and memory formation extends previous research showing better memory for information encoded during the presentation or expectation of binary (i.e., large versus small) rewards, with neural correlates overlapping those obtained in the present study[2,3,5,8].

Moreover, our results support previous studies reporting better memory for information presented in association with feedback, such as positive versus negative feedback[15,16], larger feedback prediction errors[16,21,33,34], or the unsigned feedback prediction error[34]. However, some other studies report that memory was not modulated by feedback magnitude[1,21] or feedback prediction error[35], or was negatively influenced by the feedback prediction error[36]. Unfortunately, these studies, although being just a handful, present a large variety in experimental parameters which may have contributed to these seemingly discrepant results. To name a few: encoding type (incidental versus intentional), memory type (associative versus recognition memory), encoding-testing delay period (short versus long), sleep during the delay period (with versus without sleep), task relevance of memoranda, timing of stimuli, and reinforcement learning paradigm (Pavlovian versus instrumental). While the potential influence of these factors on reward-related memory have been discussed elsewhere[16,35], we note that many of the studies reporting a positive impact of feedback value on memory formation (including ours), presented the information to be remembered (i.e., the memoranda) in close temporal vicinity to the feedback[16,21,33,34], while those reporting no or a negative impact of feedback value presented the memoranda prior to the

feedback[35,36]. For example, Jang et al.[35] presented rewards either before memoranda (to induce reward anticipation), during memoranda (to elicit what the authors termed an "image prediction error"), or after memoranda (to elicit a feedback prediction error). Strikingly, while neither the feedback prediction error nor the reward anticipation impacted on subsequent memory performance, the image prediction error, elicited by the presentation of memoranda, correlated positively with subsequent memory performance[35]. Thus, reward delivery might influence memory encoding in a limited time window, perhaps constrained by the rapid phasic response of dopamine neurons to reward delivery (<500 ms;[9,10]). Yet, because a few other studies showed that activating the reward system before and after memoranda also increased subsequent memory performance[2,3], further research needs to determine to what extent memory formation depends on the relative timing between memoranda and different aspects of reward, and how these relate to the different response profiles of the dopamine system, i.e., phasic bursts and dips, tonic activity, and sustained ramping responses[14,21].

During memory encoding, activity in the VTA and brain regions implicated in memory formation (i.e., HC and PHG) was modulated by average reward in a nonlinear fashion (i.e., inverted U-shape). Memory performance (as tested in a subsequent memory test) showed the same (inverted U-shape) modulation by average reward, whereby intermediate levels of average reward yielded best memory retention. The finding that average reward modulates feedback-related neural activity in the VTA and the HC during encoding, with similar impact on subsequently tested memory performance, supports theories postulating that reward enhances memory formation via dopaminergic influences on HC plasticity[11,12]. Yet, reward feedback induces rapid phasic bursting activity of dopamine neurons[9,10], while average reward is presumably encoded by slower changes in tonic dopamine levels[17,18,37]. These seemingly discrepant findings can be reconciled by animal research showing that the overall phasic dopamine response scales with the number of "active" dopamine neurons, while the number of active dopamine neurons is reflected in tonic dopamine levels[19,38].

Notably, this interpretation suggests that the HC loci for the effects of feedback value and immediate reward on memory

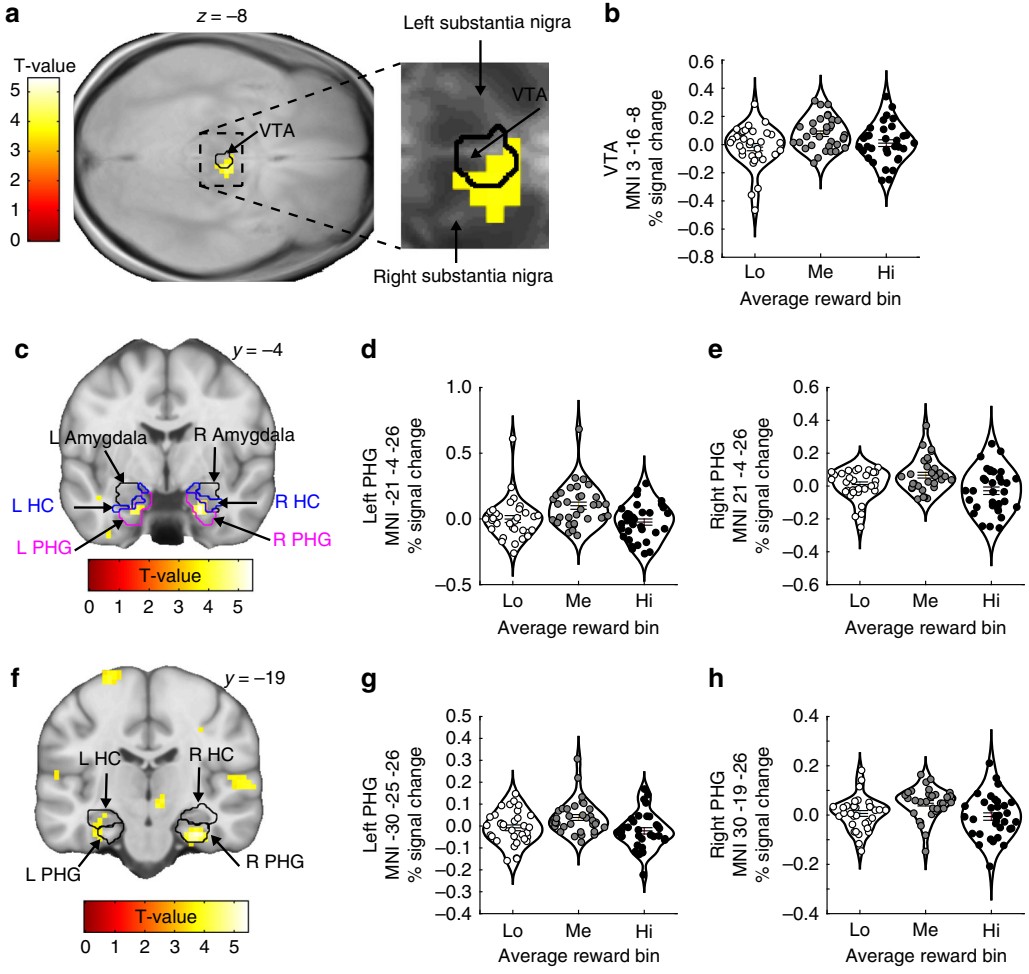

**Fig. 4 BOLD signal as a function of the average reward level.** BOLD signal in the VTA ROI (**a**, **b**), and the HC/PHG ROIs (**c–h**) was modulated in a nonlinear fashion with increasing levels of average reward. For display purposes, the violin plots show the average % signal change extracted from 3 mm spheres centered on peak voxel coordinates for each significant activation cluster. The horizontal lines indicate mean ± SEM. The BOLD signal is shown using an uncorrected threshold of $p < 0.001$. VTA ventral tegmental area ROI, HC hippocampus ROI, PHG parahippocampal gyrus ROI. Amygdala amygdala ROI. Source data are provided as a Source Data file. T-statistics were obtained from $t$-tests.

**Table 3 BOLD signal correlating non-linearly (inverted U-shape) with changes in average reward levels in a priori ROIs and the amygdala ROI.**

|  | Hemisphere | MNI peak coordinate | | | T(32) | $P_{FWE, SVC}$ |
| --- | --- | --- | --- | --- | --- | --- |
|  |  | x | y | z |  |  |
| Nonlinear modulation by average reward (inverted U-shape) | | | | | | |
| Reward mask | | | | | | |
| Ventral tegmental area | Right | 3 | −16 | −8 | 3.504 | 0.033 |
| Memory mask | | | | | | |
| Hippocampus/parahippocampal gyrus | Left | −30 | −25 | −26 | 4.486 | 0.024 |
| Hippocampus/parahippocampal gyrus | Left | −21 | −4 | −26 | 4.326 | 0.035 |
| Hippocampus/parahippocampal gyrus | Right | 21 | −4 | −26 | 5.435 | 0.002 |
|  |  | 27 | −16 | −29 | 5.216 | 0.004 |
|  |  | 30 | −19 | −26 | 5.193 | 0.004 |
| Amygdala ROI (post-hoc) | | | | | | |
| Amygdala[a] | Left | −21 | −4 | −26 | 4.326 | 0.011 |
| Amygdala[a] | Right | 21 | 4 | −26 | 5.435 | <0.001 |

$p_{FWE, SVC}$ indicates the p value resulting from family-wise error (FWE) small volume correction (SVC) on peak voxel activity within a priori ROIs and the amygdala ROI. T-statistics were obtained from $t$-tests.
ns, not significant.
[a]The amygdala was not part of the initial hypotheses, thus a stricter (Bonferroni-corrected) statistical threshold was applied in order to infer any involvement of the amygdala ($\alpha = 0.0167$).

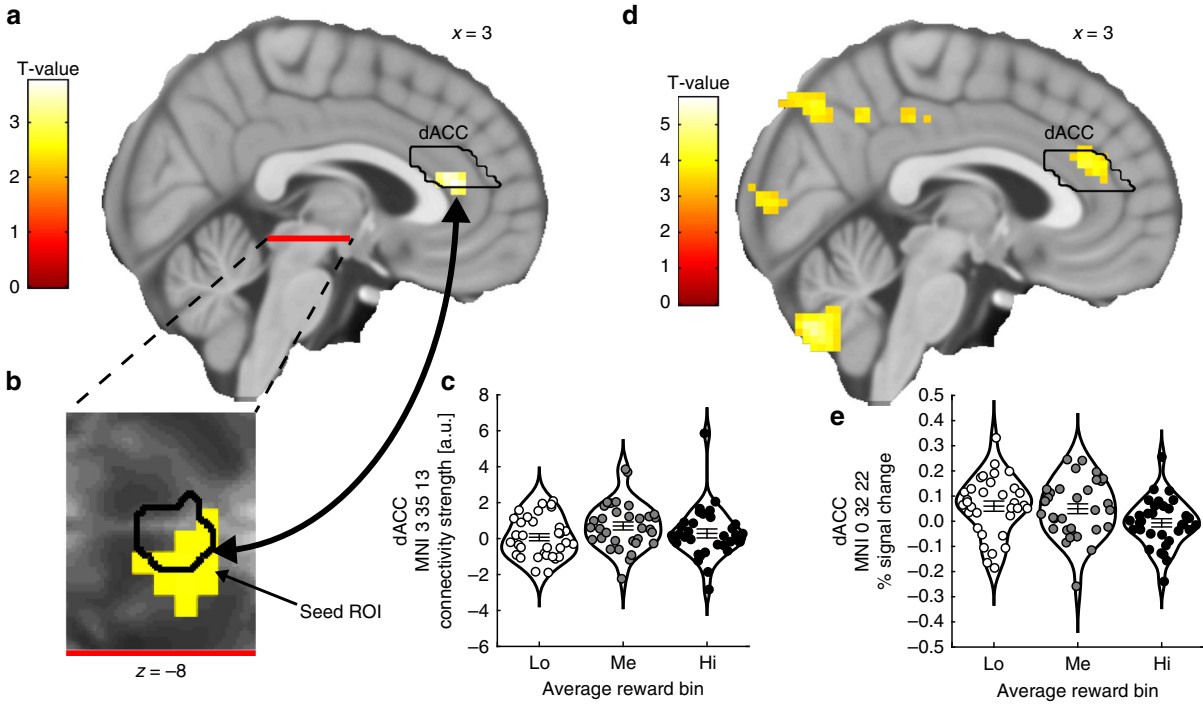

**Fig. 5 VTA–dACC functional connectivity and dACC BOLD signal as a function of the average reward level. a** Functional connectivity between the VTA seed ROI (see **b**, same figure for a zoomed-in display of the VTA seed ROI) and the dACC was nonlinearly modulated by average reward levels (see **c**, same figure for extracted beta parameters). **b** Zoomed-in display of the VTA seed ROI. **c** Nonlinear modulation (inverted U-shape) of VTA–dACC functional connectivity by average reward levels. **d**, **e** BOLD signal in the dACC decreased monotonically with average reward levels. The violin plots show the average strength of functional connectivity (**c**) and average % signal change (**e**) extracted from 3-mm spheres centered on the peak voxel coordinates within the dACC ROI in **a** and **d**, respectively. The horizontal lines indicate mean ± SEM. Activations are shown using an uncorrected threshold of $p < 0.001$. dACC dorsal anterior cingulate ROI. Source data are provided as a Source Data file. T-statistics were obtained from *t*-tests.

encoding should overlap, yet a conjunction analysis revealed no significant voxel overlap between the different contrasts. Seen through the lenses of phasic and tonic dopamine, such a result has been predicted by previous literature based on notions that different types of dopamine receptors show different degrees of affinity and are heterogeneously distributed throughout the HC. For example, Shohamy and Adcock[14] theorized that tonic dopamine acts on D5 receptors, which are mainly extrasynaptic, while phasic dopamine is restricted to engage other types of dopamine receptors located within a synapse. Moreover, Edelmann and Lessman[39] suggested that tonic dopamine firing only activates high-affinity D2 receptors, while phasic dopamine firing additionally and briefly activates low-affinity D1 and D5 receptors. In addition, Rosen et al.[40] showed that the flow of information between regions of the HC may be controlled by the dopaminergic midbrain. Specifically, optogenetic stimulation of the VTA that simulated tonic/phasic mode of dopamine release caused inhibition/facilitation of postsynaptic potentials in CA1 pyramidal neurons following Schaffer collateral stimulation, thus suggesting that information flow within the HC is determined by the current mode of dopamine activity. Because this effect was specifically dependent on D4 receptors, the impact of tonic/phasic dopamine activity might be most pronounced in regions of the HC with a higher density of D4 receptors. In summary, phasic and tonic dopamine activity may engage different regions of the HC by acting upon different dopamine receptors spread heterogeneously across the HC.

Beyond average reward, a sustained "ramping" dopaminergic response, consistent with tonic dopamine activity, can be triggered by the anticipation of uncertain rewards[20]. One study showed that memory for images presented during periods of uncertain reward anticipation was enhanced when tested after

both short and long delays[21]. Another study reported better memory, as tested after a short delay only, for items encoded during high-risk (versus low-risk) contexts, where risk was determined by the variance of the reward magnitude[34]. While the authors speculated that differences in arousal mediated the effect of risk on memory encoding, it is tempting to propose that tonic dopamine activity, induced by uncertain reward anticipation (caused by a large variance in reward magnitude), may also have contributed to this effect. Future neuroimaging studies need to elucidate whether the effects of average reward, uncertain reward anticipation, and risk on memory encoding are related to similar engagement of the VTA.

Regarding the nonlinearity of the modulation by average reward in the present study, it was previously shown in humans that slightly increased levels of dopamine (achieved via L-DOPA administration) enhanced memory formation, while excessive dopamine levels had a detrimental impact (i.e., inverted U-shape[27], see also[28]). By increasing the overall level of average reward, as compared with our previous study[16], we show that elevated levels of average reward also impede memory formation. What could explain this nonlinear influence of average reward (and dopamine) on memory formation? Dopamine is important for optimizing energy expenditure[41], and tonic dopamine may signal opportunity costs[17,18], the value of work[37], or effort[42]. Accordingly, low average reward (e.g., food scarcity during winter) promotes sloth because the value of work is low, while increasing levels of average reward (e.g., food availability during spring) promotes action by signaling an increased value of work. However, when rewards are bountiful (e.g., a domestic animal's constant access to food), the value of work is constant and hard work is unnecessary. It therefore seems unreasonable that average reward has a monotonously increasing impact on behavior.

Instead, as detailed next, we propose that the behavioral impact of average reward is first determined by the dACC, a brain region involved in the integration of different motivational factors, and then actuated via dACC–VTA interactions.

The dACC integrates different types of motivational information, such as reward, effort, and punishment[24,25], and determines behavior via top-down influences on the VTA[22,23]. The VTA is well known for its role in enabling motivated behavior[43]. A recent elaborate model suggests that dACC function can be described by two integrated modules[26]. One module monitors environmental states, such as the effort, risk, or reward associated with obtaining a goal, and selects appropriate actions, while the other module regulates the release of neuromodulators, including reward-related dopamine release from the VTA. In line with this model, we show a monotonic relationship between average reward levels and dACC BOLD (in accordance with a dACC monitoring function), and a nonlinear modulation by average reward on VTA–dACC functional connectivity (which is compatible with a dACC neuromodulatory regulation function). VTA–dACC functional connectivity was highest for the intermediate level of average reward during encoding, as was activity in the VTA and subsequently tested memory performance. Of note, dACC activity is increased by factors that reduce the expected value of future rewards, including effort needed to obtain the rewards, uncertainty associated with the reward outcome, error likelihood, and perceived decision difficulty (for reviews, see refs. [44,45]). Thus, because average reward levels estimate the current likelihood of receiving a large reward, dACC activity should be negatively correlated with average reward levels, a prediction corroborated by the present results. Given our results in combination with previous research, we propose that the dACC encodes average reward, determines its behavioral impact (after considering other concurrent factors), and then transmits this information to memory circuitry via the VTA.

The neuronal mechanisms in the VTA that are impacted by a dACC–VTA modulation are largely unknown, yet the ACC projects directly to VTA dopamine neurons via glutamatergic projections[46,47], and glutamatergic afferents to dopamine neurons supposedly control transitions between phasic and tonic activity[38,48]. Moreover, shifting the balance between excitatory glutamatergic and inhibitory GABAergic inputs to the VTA increased the spontaneous activation of dopamine neurons[49], and glutamate iontophoresis increased both the baseline firing rate and the burst firing frequency of dopamine neurons[50]. Thus, one plausible mechanism, which certainly needs empirical confirmation, is that top-down control of the VTA from the dACC occurs via glutamatergic inputs to the VTA that alter tonic dopamine activity by changing the number of spontaneously "active" dopamine neurons (i.e., neurons that may elicit phasic burst firing), and thus also the overall phasic dopamine response[19,38].

Although not part of our initial hypotheses, post-hoc tests indicated that feedback-related activation clusters (both monotonically increasing by feedback value and nonlinearly modulated by average reward) extended into the bilateral amygdala. Previous studies looking at reward-related memory enhancements have not reported the involvement of the amygdala[1,3–6], which in some cases may be due to highly constrained ROI analyses[7,8]. However, amygdala's involvement in reward-related memory encoding is not surprising, given its role in emotion-related memory enhancement. For example, the amygdala was activated bilaterally when participants viewed pleasant and unpleasant images, and the degree of amygdala activation correlated with subsequently tested memory for these images[51]. Moreover, the amygdala modulates the formation of hippocampal-dependent episodic memories[52], possibly via its ability to encode different aspects of emotionality, such as valence and salience[53]. Because the amygdala is interconnected with both the HC and the VTA[54], amygdala modulation of memory encoding in the present study could be accomplished in (at least) two different ways. First, via direct influences on hippocampal plasticity[55], or by regulating VTA dopamine neurons that project to the HC, for example by transmitting motivational salience signals to the VTA[56]. Future research should further clarify the role of the amygdala in reward-related memory enhancements.

The VTA encoded both feedback value and average reward while the NAcc only encoded feedback value. At first glance, the divergent encoding of average reward in the VTA and the NAcc is surprising given the extensive dopaminergic input from the VTA to the NAcc[57], in combination with results showing co-activation of the VTA and the NAcc in the context of prediction errors[58,59] and during rest[60]. However, the NAcc receives input from many brain regions besides the VTA[31], and may therefore be influenced by information beyond that transmitted by VTA dopamine neurons[61]. In line with this notion, one study alternated visual and auditory cues as being the relevant feature to predict upcoming rewards, and showed that the ventral striatum (of which the NAcc is part) tracked the value of the currently relevant stimulus only, and was activated by the correctness of the response rather than the outcome's value[62]. In a related study, participants learned the timing of reward outcomes rather than their magnitudes, and while the VTA tracked parameters related to both reward magnitudes and timings, the ventral striatum only tracked the reward timings, i.e., the task-relevant parameter. Similarly, participants in the present study learned the value of character–object associations, and this task-relevant feature was tracked by both the VTA and the NAcc. By contrast, the task-irrelevant average reward level was only tracked by the VTA. Thus, the ventral striatum/NAcc may update the features of the environment that are most relevant to drive behavior[63], a specific role that is not shared by the VTA.

Some limitations of the present study needs to be acknowledged. First, we show that average reward modulates the encoding of character–object associations, yet it remains unknown to what extent average reward modulates the encoding of other types of information within a trial, such as memory for a particular character identity or objects. While the present study was not designed to test memory for individual stimuli, the experimental paradigm could easily be adapted to test such aspects by, for example, replacing the associative memory test with a stimulus recognition test. Another limitation, shared by many studies on human reward-related memory modulation, is that memory was tested after a relatively short delay period (i.e., 20 min after encoding in the present study). Thus, it remains unknown whether and how feedback value and average reward in the present study may modulate subsequent dopamine-dependent consolidation processes. Dopamine is believed to enable consolidation of long-term potentiation (LTP)[11], a cellular model of long-term memory[64]. In brief, LTP posits that memories are initially encoded by changes in synaptic strengths following synaptic activation (early LTP), and that these changes, and thus also the memories they encode, quickly vanish unless the synapses undergo a stabilization/consolidation process that involves the synthesis of plasticity-related proteins (late LTP)[11,12]. While experimental evidence suggests that dopamine is important for both early and late LTP, it may be particularly relevant for late LTP. For example, administering dopamine antagonists before learning in a one-trial reward learning task impeded memory when tested after 24 h, but not when tested after 30 min[65,66]. Accordingly, the impact of dopamine-releasing events during encoding, such as reward, may be particularly evident when memory is tested after long delays (i.e., after late LTP has occurred). Some human research support this notion by showing

that reward-related enhancements of memory emerged only when tested after long delays (>24 h)[2,5,21] or after a nap[4] (which may allow for dopamine-related consolidation processes[67,68]). Yet, other studies report reward-related memory modulations when tested both after short (<30 min) and long delays[16,35], or when tested after short delays only[1,15,34]. The latter results suggest that reward-related memory modulations do not always pertain to consolidation processes, but may also act directly on the encoding of a memory, a notion which is in line with the present results. Yet, future studies need to address the impact of average reward on later consolidation processes. A final limitation is that the current fMRI approach cannot address what low-level mechanism(s) could explain reward influences on memory encoding. Some experimental evidence suggests that dopamine lowers thresholds for LTP induction[69]. Another interesting option relates to the notion of a three-factor rule of synaptic plasticity[70]. In brief, this theory posits that co-activation of pre- and postsynaptic neurons sets an eligibility trace that allows synaptic change, but only in the presence of a third modulating factor. This modulatory factor may be dopamine, but could also be any other neuromodulator known to impact learning, such as acetylcholine, noreprinephrine, or serotonin. This rule explains rapid behavioral change without the need for consolidation, and is supported by recent experimental evidence obtained in the striatum, the prefrontal and visual cortices, and in the HC (see Gerstner et al.[70] for a recent review). Other mechanisms have also been suggested, i.e., see Box 5 of Lisman et al.[11] and Shohamy and Adcock[14].

To summarize, we show that reward feedback exhibits (1) an immediate and positive effect on memory encoding that engages the NAcc, the VTA, and hippocampal brain regions, and (2) a more sustained non-linear effect via its integration into average reward levels, which in turn modulates neural responses within the VTA–HC loop, as determined by the dACC and its interactions with the VTA. These results demonstrate that the accumulation of reward contributes to memory encoding success, with intermediate levels of average reward providing the most beneficial setting for memory encoding. The impact of average reward across cognitive domains, and its interaction with inter-individual differences related to dopamine functionality, remain to be investigated.

## Methods

Functional MRI data were acquired while participants performed an associative memory task consisting of one memory encoding session and one memory test session. We designed different computational models explaining how immediate feedback value and average reward might differentially impact memory formation, and then fitted them to memory performance during the test session. The parameters derived from the most parsimonious model were then combined with fMRI data to uncover brain activity that correlated with the different dimensions of reward delivery during memory encoding. Specific details about the experimental procedures are presented below.

**Participants**. After having provided written and informed consent according to the ethical regulations of the Geneva University Hospital, 34 participants joined the experiment. All participants were right-handed, native French speakers, and without any previous history of psychiatric or neurological disorders. The study was performed in accordance with the Declaration of Helsinki. Data from one participant had to be excluded due to him/her falling asleep during the memory test phase. Thus, data from the remaining 33 participants were included in the subsequent analyses (15 females; average age ± SEM: 25.242 ± 0.874).

**Associative memory task**. Each participant performed an associative memory task consisting of one memory encoding session (9 min) and one memory test session (9 min). Both sessions were performed during fMRI scanning and were separated by 20 min.

Memory encoding session: In each trial during the encoding session, a fixation cross was first presented for 3 s, followed by the face of a cartoon character presented together with a pair of objects[71] for 3.5 s (Fig. 1a). Participants were instructed that the character moderately liked or very much liked one object in

each pair, while disliking the other object. Participants had to indicate which object they thought that the character liked (moderately or very much) by pressing one of two buttons with the right hand. A feedback display, consisting of a number enclosed by a colored circle was then presented for 1 s (Fig. 1b). The feedback indicated whether the character moderately liked (+1, blue circle), very much liked (+5, green circle), or disliked (−1, magenta circle) the selected object (Fig. 1c). Besides informing about a preference, the feedback also indicated how many points each selection was worth. The feedback was presented 4 s after the onset of the face/ objects display. Participants were instructed that they could gain additional points by correctly remembering each character–object preference for a subsequent memory test phase (see below), and that the points would be converted into a monetary bonus at the end of the experiment.

Critically, the feedback was manipulated to yield different average reward levels for the six different characters presented during the memory encoding session. Specifically, while all characters received four −1 feedbacks, two characters associated with high (Hi) average reward received one +1 feedback and five +5 feedbacks, two characters associated with medium (Me) average reward received three +1 feedbacks and three +5 feedbacks, while two characters associated with low (Lo) average reward received five +1 feedbacks and only one +5 feedback (Fig. 1d). Each of the six characters was presented together with each of ten possible object pairs (i.e., the object pairs were constant across the experiment and participants), for a total of 60 trials. All ten character–object associations were encoded for one character (i.e., ten trials), before the same object pairs were presented together with the next character, and so forth for all six characters. For each participant, the characters were randomly assigned to one of the different average reward levels (i.e., either Lo, Me, or Hi).

Test session: During the test session, participants were presented with the same characters and same object pairs, and were instructed to recall and select the character's preferred object in each object pair. Selecting an object previously associated with positive feedback (i.e., +1 or +5) earned the corresponding points (i.e., +1 and +5), while rejecting an object associated with −1 feedback earned +1 points. Note that correctly remembering associations encoded during +1 feedback or those encoded during −1 feedback (correct rejection) would yield +1 point in both cases. Participants lost 1 point if they selected a disliked object, i.e., either by selecting a previously disliked object or by not selecting a previously rewarded object. Testing trials were identical to encoding trials (Fig. 1a), except that no feedback was provided. The order of characters, object pairs, and the side of object presentation within a pair, were pseudorandomized. First, each character (each with the 10 corresponding object pairs, i.e., 10 trials) was presented once before any other character was repeated. Second, the order of the ten object pairs was randomized for each character. Third, the side of object presentation within a pair was reversed (with respect to the side of presentation during the encoding session) for half of the object pairs for each character.

Procedure: To get familiar with the task, participants first trained the task outside the MRI scanner. Six character–object associations were encoded for one character and then tested after a delay of ~5 min. Neither the character nor the trained object pairs were subsequently used during the main experiment. Participants then performed the associative memory task inside the MRI scanner, as described previously (i.e., one encoding session and one test session).

Statistical analysis: Memory performance during the test session was calculated as the proportion of correct responses: i.e., selection of objects that had been rewarded for a given character during the encoding phase (+1, +5 feedbacks), or rejections of objects classified as "disliked" by a given character (−1 feedback). We computed the number of correct responses for each combination of immediate feedback value (+1, +5, and −1) and average reward (Lo, Me, and Hi).

The data were analyzed using linear mixed models of the following form: $\mathbf{Y}_i = \mathbf{X}_i\mathbf{B} + \mathbf{Z}_i\mathbf{b}_i + \mathbf{e}_i$, where $\mathbf{Y}_i$ represents a vector of values of the dependent measure of interest for the $i$th participant, $\mathbf{X}_i$ represents a matrix of $p$ predictors (independent variables) for the $i$th participant, $\mathbf{B}$ represents a vector of $p$ fixed effect beta weight estimates for each predictor in $\mathbf{X}_i$, $\mathbf{Z}_i$ represents a matrix of $q$ random effect predictors, $\mathbf{b}_i$ represents a vector of $q$ random effect estimates, and $\mathbf{e}_i$ represents a vector of the model fit error, representing the discrepancy between the model prediction for each observation from the $i$th participant and the actual value of that observation. For behavior and modeling data, two categorical predictors with three levels each were used: Feedback value (−1, +1, and +5) and Average reward (Lo, Me, and Hi) along with their interaction. Moreover, each participant was treated as a random variable, i.e., the matrix $\mathbf{Z}$ contained one column for each participant pertaining to that participant's random effect estimate $\mathbf{b}$.

ANOVA was initially used to analyze the results, and significant effects were further investigated using two-tailed paired $t$-tests. Of note, while linear mixed models are in many ways superior compared with alternative approaches[72], the calculation of standard effect sizes for these models is still heavily debated[73]. For this reason, we restricted the reporting of standardized effects sizes to the difference between means, which was calculated using Cohen's $d$:

$$d = \frac{\mu_1 - \mu_2}{\sqrt{(\sigma_1^2 + \sigma_2^2)/2}}.$$

**Computational approach**. To provide a more fine-grained (i.e., trial-wise) analysis of behavior and fMRI data, we created and confronted different computational models for successful memory formation depending on different types of reward

available at the time of encoding. As such, the average reward $\bar{r}$ is calculated using an exponential running average[74] while the feedback values (fb) are the actual reward values (i.e., −1, +1, or +5):

$$\bar{r}(t) = v \times \mathrm{fb}(t) + (1 - v) \times \bar{r}(t - 1),$$

where $v$ is a learning rate which determines the integration rate of recent rewards into the overall estimate of current average reward. The probability $p_E$ of successfully encoding information presented in a trial $t$ is described by a logistic function:

$$p_E(t) = \frac{1}{1 + e^{-R(t)}},$$

$R(t)$ is defined in different models based on previous data linking reward and dopamine to memory formation, as described next.

A "null"-model assumes no influence of reward on memory performance:

$$R_{\mathrm{null}}(t) = C_0.$$

A second model accounts for the impact of feedback value on memory formation ($C_{\mathrm{fb}}$):

$$R_{\bar{r}}(t) = C_0 + C_{\mathrm{fb}} \times \mathrm{fb}(t).$$

A third model accounts for the hypothesized impact of average reward on memory formation ($C_{\bar{r}}$):

$$R_{\bar{r}}(t) = C_0 + C_{\bar{r}} \times \bar{r}(t).$$

A fourth model accounts for independent additive contributions from reward magnitude and average reward:

$$R_{\mathrm{fb}+\bar{r}}(t) = C_0 + C_{\mathrm{fb}} \times \mathrm{fb}(t) + C_{\bar{r}} \times \bar{r}(t).$$

A fifth model also includes their interaction ($C_{\mathrm{fb}\bar{r}}$):

$$R_{\mathrm{fb}+\bar{r}+\mathrm{fb}\bar{r}}(t) = C_0 + C_{\mathrm{fb}} \times \mathrm{fb}(t) + C_{\bar{r}} \times \bar{r}(t) + C_{\mathrm{fb}\bar{r}} \times \mathrm{fb}(t) \times \bar{r}(t).$$

A sixth model accounts for the nonlinear modulation of memory formation by increasing average reward ($C_{\bar{r}^2}$), inspired by the notion that average reward is encoded by tonic dopamine[17,18] and the nonlinear modulation of episodic memory formation by increased dopamine levels:[27]

$$R_{\mathrm{fb}+\bar{r}^2}(t) = C_0 + C_{\mathrm{fb}} \times r(t) + C_{\bar{r}^2} \times (\bar{r}(t) - w))^2,$$

where $w$ indicates the "optimal" level of average reward, i.e., the point where lower or higher levels of average reward are detrimental to memory encoding.

A seventh and final model is similar to the sixth model, but presumes that average reward levels are calculated for each cartoon character independently. To account for this possibility, an initial level of average reward ($C_{\bar{r}0}$) was therefore fitted across cartoon characters.

$C_{\mathrm{fb}}$, $C_{\bar{r}}$, $C_{\mathrm{fb}\bar{r}}$, and $C_{\bar{r}^2}$ regulate the modulation of overall reward levels by reward magnitude, average reward, their interaction, and the nonlinear modulation by average reward, respectively. The free parameters $C_0$, $C_{\mathrm{fb}}$, $C_{\bar{r}}$, $C_{\mathrm{fb}\bar{r}}$, $C_{\bar{r}^2}$, $v$, $w$ were fitted to each participant's data through maximum likelihood estimation, i.e., by minimizing the negative log-likelihood estimation function (LLE) for logistic regression:

$$\mathrm{LLE} = -\sum_{t=1}^{n} y(t) \times \log p_E(t) + (1 - y(t)) \times \log(1 - p_E(t)),$$

where $y(t)$ is the observed outcome (i.e., hit/miss) in each trial $t$. To validate and confront these models, their fit to behavioral data were compared using AIC[75] which accounts for different numbers of fitted variables:

$$\mathrm{AIC} = 2 \times k + 2 \times \mathrm{LLE}.$$

**MRI data**. Image acquisition: MRI images were acquired using a 3T whole body MRI scanner (Trio TIM, Siemens, Germany) with a 12-channel head coil. Standard structural images were acquired with a T1 weighted 3D sequence (MPRAGE, TR/TI/TE = 1900/900/2.27 ms, flip angle = 9°, voxel dimensions = 1 mm isotropic, $256 \times 256 \times 192$ voxels). Proton density (PD) structural images were acquired with a turbo spin echo sequence (TR/TE = 6000/8.4 ms, flip angle = 149°, voxel dimensions = $0.8 \times 0.8 \times 3$ mm, $205 \times 205 \times 60$ voxels). The PD scan was used to confirm the location of VTA activation, as it allows the identification of the substantia nigra, a brain region located just laterally to the VTA[58]. The acquisition volume was oriented in order to scan the brain from the lower part of the pons to the top of the thalamus. Functional images were acquired with a susceptibility weighted EPI sequence (TR/TE = 2100/30 ms, flip angle = 80°, voxel dimensions = 3.2 mm isotropic, $64 \times 64 \times 36$ voxels).

MRI data analysis: Functional MRI data were preprocessed and then analyzed using the general linear model for event-related designs in SPM8 (Welcome Department of Imaging Neuroscience, London, UK; http://www.fil.ion.ucl.ac.uk/spm). During preprocessing, all functional volumes were realigned to the mean image, co-registered to the structural T1 image, corrected for slice timing, normalized to the MNI EPI-template (via a 12 parameter affine transformation model with trilinear interpolation), and smoothed using an 8 mm FWHM

Gaussian kernel. Statistical analyses were performed on a voxelwise basis across the whole-brain. At the first-level analysis, individual events were modeled by a standard synthetic hemodynamic response function (HRF) and six rigid-body realignment parameters were included as nuisance covariates when estimating statistical maps. Contrasts (see below) were then calculated and the contrast images entered into second-level $t$ tests implemented in SPM.

Parametric modulation by model-derived reward parameters: By fitting computational models to memory performance from the test session, we obtained parameters that best explained how changes in feedback value and average reward during memory encoding influenced subsequent memory performance[16]. Combining fitted model parameters with fMRI data, we could then trace back the neural correlates of those distinct dimensions of reward (that we manipulated in the task) during associative memory encoding.

To this end, we created an event-related design that included three event-types modeled as stick functions (i.e., with a 0 duration) that were respectively time-locked to the onset of the display of character/object pairs, the button press of the response, and the feedback in each trial. We then added trial-by-trial estimates of model-derived reward parameters, i.e., feedback values (i.e., −1, +1, and +5) and average reward levels (calculated as an exponential running average of feedback values), as parametric modulators to the feedback onset times. To disentangle neural activity related to different parametric modulators, the vectors containing respective parametric modulator were orthogonalized. This process allows studying the neural correlates of one parametric modulator independently of another[76].

Functional connectivity analysis of VTA–ACC interactions: For the connectivity analysis, we used the gPPI approach which has the benefit of accommodating multiple task conditions, including parametric modulators, in the same connectivity model[77]. Physiological variables were created by extracting the deconvolved times series from each seed ROI. Psychological regressors were created by convolving each onset regressor and parametric modulator with the canonical HRF. Psychophysiological interaction (PPI) terms (which allow the identification of voxels showing task-dependent covariation with the seed ROIs) were created by multiplying the time series from the psychological regressors with the physiological variable. All of the above was performed for each participant separately, and individual gPPI models were created by including the corresponding physiological variables, the psychological regressors, and the PPI terms.

Regions of interest (ROIs): The selection of a priori ROIs used for small volume corrections (SVCs) was guided by previous literature on reward-related memory enhancements[3,5], reporting the involvement of memory-related brain structures such as the HC and the PHG, as well as structures coding for reward, such as the NAcc, the VTA, and the dACC.

ROIs for the NAcc, HC, and the PHG were obtained from the WFU toolbox[78]. The VTA ROI was obtained from a recently published probabilistic atlas of the VTA, by including voxels identified as being part of the VTA in 50% of participants[60]. The dACC ROI was defined by limiting the ACC ROI of the WFU toolbox to $z > +12$ and $y > +16$[79]. A "reward mask" was created by combining the NAcc and VTA ROIs, while a "memory mask" was created by combining the HC and the PHG ROIs. Both of these masks were used to test the parametric modulation by feedback value and average reward. By contrast, the dACC ROI was only used to test specific hypotheses related to VTA–dACC interactions. For visualization purposes, an amygdala ROI obtained from the WFU toolbox was included in some of the figures.

Additional post-hoc analyses tested the involvement of the amygdala by using SVCs on a ROI mask of the bilateral amygdala obtained from the SPM Anatomy Toolbox Version 3.0 (all sub-regions of the amygdala included in the mask)[80].

**Statistical analyses**. The obtained results are displayed using a threshold of $p < 0.001$ and a minimum cluster size of ten contiguous voxels, unless otherwise reported. SVCs using a threshold of $p < 0.05$ family-wise error rate (FWER) for multiple comparisons were obtained using a priori ROI masks reported above. Bonferroni-corrected statistical thresholds were applied to two post-hoc tests that investigated the involvement of the amygdala. Specifically, because two post-hoc tests were performed, the statistical threshold was corrected to 0.017 (0.05/3) for these tests. Conditions were compared using paired $t$-tests, as implemented in SPM.

For display purposes, the average brain activity were extracted from 3-mm spheres centered on peak voxels within significant clusters of activation and shown as insets. While the feedback values were constant (−1, +1, and +5), average reward levels changed continuously between trials. For this reason, we calculated and displayed the average brain activity within three equally sized bins of average reward levels, i.e., the 0–33rd percentiles, 34–66th percentiles, and 67–100th percentiles.

**Reporting summary**. Further information on research design is available in the Nature Research Reporting Summary linked to this article.

## Data availability

The data used to produce the results reported in the manuscript can be made available upon appropriate request. In addition, the source data underlying Figs. 1e, f, 2e, f, g, 3b, c,

e, f, h, i, 4b, d, e, g, h, 5c, e, and Supplementary Figs. 1b and 2b are provided as a Source Data file. A reporting summary for this Article is available as a Supplementary Information file.

## Code availability

Data code used to produce the results reported in the manuscript can be made available upon appropriate request.

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

## Acknowledgements

This study was conducted on the imaging platform at the Brain and Behavior Lab (BBL) and benefited from support of the BBL technical staff. This work was supported by the National Center of Competence in Research (NCCR) Affective Sciences financed by the Swiss National Science Foundation (grant number: 51NF40-104897) and hosted by the University of Geneva, and the Swiss National Science Foundation (Grant numbers: 320030-159862 and 320030-135653).

## Author contributions

K.C.A. designed the experiment, collected and analyzed the data, and wrote the manuscript. E.E.K. collected and analyzed the data. S.S. designed the experiment, analyzed the data, and wrote the manuscript.

## Competing interests

The authors declare no competing interests.
