## [Peer Review File · Nature Communications]

Reviewers' comments:

Reviewer #1 (Remarks to the Author):

Aberg and colleagues present a manuscript detailing the influence of more phasic-like and tonic-like reward signals on subsequent episodic memory. Participants completed a reward learning task during the collection of fMRI, in which they could either earn or lose a small reward (\$1) or earn a large reward (\$5) in response to decision-making. With their paradigm, authors were able to see how immediate reward feedback influenced memory, as well as how the average award influences memory. Behaviorally, the found linear increases in reward feedback and subsequent memory, and found an inverted U relationship between average reward signal and subsequent memory. Neuroimaging data showed that ventral striatal and hippocampal activity showed linear responses akin to immediate reward feedback, whereas VTA and hippocampus showed non-linear responses akin to the average reward signal. Further, functional coupling between the VTA and the dACC showed similar non-linear responses. The question of how immediate versus average reward values influence memory is highly timely. Further, the authors use an elegant design complimented with a nice computational approach. The results are compelling, and add to a growing literature on reward feedback and memory. There were a few weaknesses in the paper, centered on relating this work to prior work and certain analyses procedures, that derail the overall impact of the paper. These concerns and others are detailed below.

1. Many of the influences of dopamine in the rodent literature, in particular tonic dopamine, has been shown to influence memory consolidation rather than encoding. However, the authors only test memory after a 20 minute delay. Further discussion of how the current results relate to the underlying behavioral neuroscience studies is warranted.
2. While the introduction does a commendable job motivating the studies, there a few studies looking at reward feedback and how it influences subsequent memory which are not discussed. The authors need to relate to current findings with these prior literatures, some of which report conflicting results. Specifically, the authors should discuss Rouhani et al. (2018), Jang et al. (2019), and Stanek et al. (2019).
3. I was a little unclear about the randomization of average reward values across characters. Was this factor randomized across individual participants, or where the same reward values used for a given character for each participant. If there was no randomization, there could be confounds of character identity. Also did the authors account for any differences in interference of accumulating reward information across a given identity, or testing-order effects across identities. Supplemental analyses on these important factors should be including.
4. Modeling is not my specific expertise, but I was unclear whether average reward values was modeled for each character separately or if it was run as a continuous value across the experiment. I believe it was the latter. If so, I think the authors should also test a model in which each character identity starts with a given baseline and is updated separately. Specifically, participants could be forming higher-order representations (i.e., latent causes) in which average reward value is segmented by character identity. If

so, this could have consistent, albeit subtle, differences on the relationship between average reward and subsequent memory/neural activation.

5. An 8 mm smoothing kernel is quite large for the VTA, and I am worried that the authors could also be incorporating signal from surrounding midbrain structures (i.e., SN, red nucleus). Do the results remain the same using a smaller kernel (i.e., 4 mm) or a more ROI based approach without smoothing.

6. In regards to the bar graphs reporting the functional activations, I think rather than illustrating the peak the presentation of results should be extracted from the entire cluster. Further, statistics should be run on these peak extracted clusters as this results in a circular analysis. Rather the plots should just be used for visualization purposes (see Kriegeskorte et al., 2010 for discussion).

7. More details on the anatomical specificity of the hippocampal activation would be helpful. It is overlapping across analyses? Is it in more anterior or posterior portions. In a few samples it seems like the peaks are in cortical MTL rather than hippocampus. Perhaps presenting the data on a sagittal view (either in the manuscript or supplementals) would provide a better representation of the anatomical locations.

Reviewer #2 (Remarks to the Author):

In this study, authors used computational modeling and fMRI to test the behavioral and neural modulations by immediate and average reward amount during memory encoding, and the relationship between the two types of modulation. The manuscript is very clearly written and clever experimental manipulations were adopted to distinguish encoding modulation by immediate and average reward amount. Dissociations between so-called phasic and tonic DA activities and their distinct contribution to memory encoding is not terribly new, but showing them in one simple empirical paradigm, along with the interaction of DA system with dACC, makes a new contribution to the literature. I have a few concerns and questions that I feel needed to be addressed before this work is published.

1. I have a few questions about the way fMRI data was modeled and the rationale behind it. If I understood correctly, the onset of stimuli and feedback display were modeled in each trial, and two parametric regressors (model-derived estimates of immediate reward and average reward levels) were added to 'feedback onset'.

1a. Most simply, how were the events modeled? Was a stick function or a boxcar function was used? The term "event-related design" in the fMRI analysis description implies that a stick function was used, but it needs to be explicitly described.

1b. Why was 'response' not modeled? This would leave response-related variations of the BOLD signal unaccounted for, possibly adding noise to the results. I understand that response was not an event of interest for any analyses in this study, but it still needs to be included in the model.

1c. This last point is of more importance and applies broadly to other parts of the manuscript (including Introduction and interpretations). In essence, my point is about the assumption and approaches about when/how immediate and average reward amount would influence memory encoding. Encoding of a specific event starts with the onset of the event (or, stimulus display in the experiment) and the continuing encoding process gets influenced by feedbacks. Critically, where the action occurs in this line of process (starting from the stimulus display and continuing till feedback, or the end of the event) may differ for immediate and average reward. For example, whereas immediate reward feedback is likely to affect encoding of the event that just happened, average reward level is more likely to affect encoding of the upcoming event.

In the present study, the GLM included parametric regressors at the onset of feedback display, which will get at “whether there is any additional activation modulated as a function of the parametric regressors (i.e., linear effects of immediate reward and inverted-U shape effects of avg reward) when the average activation during feedback display is accounted for”. This makes sense for the effects of immediate reward on DA system and memory encoding, however, when it comes to the effects of average reward level, there would be at least 2 different processes to consider. Firstly, when information comes in (stimuli presentation), the motivational state, or tonic/sustained DA activation is likely to impact encoding of this information. Then at the time of feedback onset, the representation of average reward level gets adjusted – by integrating the running reward average up to that moment and the reward amount in the feedback that was just received. Given the way data were modeled, the present results likely reflect the latter, and it makes total sense that the VTA activation during this feedback integration phase is tightly linked with dACC activation. This in itself is a meaningful finding, but it does not address exactly what the study is proposing to test. When it comes to the effects of average reward level on memory encoding, the action is most likely to be happening at the onset of stimuli display.

2. The point 1c above applies to modeling of behavioral data as well. It was surprising to see that the influence of immediate vs. average reward level on memory encoding was independent of each other. In fact, again, I suspect that it may be because of the way data were modeled. Encoding of a given trial would most influenced by the immediate reward of that trial and the running average reward up to that stimulus display (rather than average reward update following the feedback). It would make more sense to test whether the average reward level activated by the onset of stimulus display (a character that is assigned to Lo, Me, or Hi average reward level) modulates the effects of immediate reward amount to memory encoding (interaction).

If this alternative model also reveals independent influence of the two on encoding, it would be important to discuss it in depth.

3. Very closely related to the point above (1c & 2) in the Introduction, authors bring up two possible scenarios (not necessarily against each other) on the sustained effects of reward on memory encoding – interactions between tonic and phasic dopaminergic activities, and top-down control of VTA by dACC.

Results clearly support the latter, but not much is discussed as to what the results mean with regards to the tonic-phasic DA interaction story. Does the lack of interaction effects on memory encoding rule out this possibility? Applying the a parametric regressor of average reward level to the onset of stimulus display (the average before being updated with the current trial reward amount) may reveal meaningful interactions between tonic and phasic DA interactions.

4. Ventral striatum is a main target of VTA DA projection, and it is surprising to see completely different reward-related response profiles in vSTR and VTA. Results suggest that VTA is NOT sensitive to immediate reward amount, and instead its activation changes in an inverted-u shaped function of average reward update - with the greatest activation increases at medium level average reward condition. vSTR, which gets downstream signal from VTA, however, is NOT sensitive to the average reward level, but instead exhibits greater activation in response to greater immediate feedback. I believe this is worth a discussion point - what are authors' thoughts on this?

5. A proper group-level inference of the model fit for behavioral results would help succinct presentation of findings. Table 1 shows an example participant's data & Figure 2E shows the model estimated values – which resembles the pattern in the actual data (Figure 1F, G). A more succinct and convincing way to draw a group-level inference would be to use a mixed-model to fit the data from all subjects in one model while controlling for variability from subjects.

6. It was a bit unclear how ROIs were used in analyses with the two different parametric regressors. Were both the linear and non-linear parametric regressors effects tested in all ROIs? Or were particular sets of ROIs were tested for each parametric regressor?

7. There is an inconsistency in terminology for NAcc, or ventral striatum. NAcc is not mentioned until Discussion – My guess is that authors are referring to the vSTR as NAcc there (or the other way around). NAcc is included in vSTR, but they are not the same, therefore should not be referred to interchangeably. Which one was the actual ROI?

8. Typos [Line 127]: “(three +1’s, one +5’s)” – Isn’t this supposed be “(three +1’s, three +5’s)”?.

Reviewers' comments:

Reviewer #1 (Remarks to the Author):

Aberg and colleagues present a manuscript detailing the influence of more phasic-like and tonic-like reward signals on subsequent episodic memory. Participants completed a reward learning task during the collection of fMRI, in which they could either earn or lose a small reward (\$1) or earn a large reward (\$5) in response to decision-making. With their paradigm, authors were able to see how immediate reward feedback influenced memory, as well as how the average award influences memory. Behaviorally, the found linear increases in reward feedback and subsequent memory, and found an inverted U relationship between average reward signal and subsequent memory. Neuroimaging data showed that ventral striatal and hippocampal activity showed linear responses akin to immediate reward feedback, whereas VTA and hippocampus showed non-linear responses akin to the average reward signal. Further, functional coupling between the VTA and the dACC showed similar non-linear responses. The question of how immediate versus average reward values influence memory is highly timely. Further, the authors use an elegant design complimented with a nice computational approach. The results are compelling, and add to a growing literature on reward feedback and memory. There were a few weaknesses in the paper, centered on relating this work to prior work and certain analyses procedures, that derail the overall impact of the paper. These concerns and others are detailed below.

1. Many of the influences of dopamine in the rodent literature, in particular tonic dopamine, has been shown to influence memory consolidation rather than encoding. However, the authors only test memory after a 20 minute delay. Further discussion of how the current results relate to the underlying behavioral neuroscience studies is warranted.

Thank you for highlighting this very relevant point. Indeed, testing memory after 20 minutes is a limiting factor for the present study, which we now acknowledge in a new Limitations section of the discussion. Specifically, we now emphasize the potential influence of dopamine-dependent consolidation processes on reward-related memory modulations by i) referencing relevant reviews of dopamine-dependent consolidation, ii) elaborating upon consolidation processes in the context of long-term potentiation (LTP), and iii) performing a mini-review of behavioral results obtained in humans. See lines 670-692 of the main text:

Another limitation of the present study, a limitation shared by many studies on human reward-related memory modulation, is that memory was tested after a relatively short delay period (i.e. 20 minutes after encoding in the present study). Thus, it remains unknown whether and how feedback value and average reward in the present study may modulate subsequent dopamine-dependent consolidation processes. Dopamine is believed to enable consolidation of long-term potentiation (LTP)¹, a cellular model of long-term memory². In brief, LTP posits that memories are initially encoded by changes in synaptic strengths following synaptic activation (early LTP), and that these changes, and thus also the memories they encode, quickly vanish unless the synapses undergo a stabilization/consolidation process that involves the synthesis of plasticity-related proteins (late LTP)^{1,3,4}. While experimental evidence suggests that dopamine is important for both early and late LTP, it may be particularly relevant for late LTP. For example, administering dopamine antagonists before learning in a one-trial reward learning task impeded memory when tested after 24 hours, but not when tested after 30 minutes^{5,6}. Accordingly, the impact of dopamine-releasing events during encoding, such as reward, may be particularly evident when

memory is tested after long delays (i.e. after late LTP has occurred). Some human research support this notion by showing that reward-related enhancements of memory emerged only when tested after long delays (>24 hours)^{7, 8, 9, 10} or after a nap¹¹ (which may allow for dopamine-related consolidation processes^{12, 13}). Yet, other studies report reward-related memory modulations when tested both after short (<30 minutes) and long delays^{14, 15}, or when tested after short delays only^{16, 17, 18}. The latter results suggest that reward-related memory modulations do not always pertain to consolidation processes, but may also act directly on the encoding of a memory, a notion which is in-line with the present results. Yet, future studies need to address the impact of average reward on later consolidation processes.

2. While the introduction does a commendable job motivating the studies, there are a few studies looking at reward feedback and how it influences subsequent memory which are not discussed. The authors need to relate to current findings with these prior literatures, some of which report conflicting results. Specifically, the authors should discuss Rouhani et al. (2018), Jang et al. (2019), and Stanek et al. (2019).

Thank you for drawing our attention to these highly relevant studies. We now mention the study by Stanek et al. in the Introduction (lines 61-64):

Related to this notion, one elegant study reported that the anticipation of uncertain rewards, a condition which is known to evoke a sustained “ramping response” of dopamine neurons¹⁹, increased incidental memory encoding for images presented during the reward anticipation period⁹.

Moreover, we now also provide a detailed discussion on how these (and other) studies relate to the present results in a new section of the Discussion:

Relating the present results to previous studies on reward-related modulations of human declarative memory formation

Beyond average reward, a sustained “ramping” dopaminergic response, consistent with tonic dopamine activity, can be evoked by the anticipation of uncertain rewards¹⁹. One study showed that memory for images presented during periods of uncertain reward anticipation was enhanced when tested after both short and long delays⁹. Another study reported better memory, as tested after a short delay only, for items encoded during high-risk (versus low-risk) contexts, where risk was determined by the variance of the reward magnitude¹⁸. While the authors speculated that differences in arousal mediated the effect of risk on memory encoding, it is tempting to propose that tonic dopamine activity, induced by uncertain reward anticipation (caused by a large variance in reward magnitude), may also have contributed to this effect. Future neuroimaging studies are needed to elucidate whether the effects of average reward, uncertain reward anticipation and risk on memory encoding are related to similar engagement of the VTA.

In the present study, the feedback value influenced memory encoding monotonically and positively, a finding that corroborates previous studies showing better memory for information presented in association with feedback, such as positive versus negative feedback^{14, 17}, larger feedback prediction errors^{9, 14, 18, 20}, or the unsigned feedback prediction error¹⁸. However, other studies reported that memory was not modulated by feedback magnitude^{9, 16} or feedback prediction error¹⁵, as well as a negative influence on memory encoding exerted by the feedback

prediction error²¹. Unfortunately, this handful of studies presents a large variety in experimental parameters, which may have contributed to these seemingly discrepant results. To name a few: encoding type (incidental versus intentional), memory type (associative versus recognition memory), encoding-testing delay period (short versus long), sleep, task-relevance of memoranda, timing of stimuli, and reinforcement learning paradigm (Pavlovian versus instrumental). While the potential influence of these factors on reward-related memory have already been discussed elsewhere^{14,15}, one interesting observation is that many of the studies reporting a positive impact of feedback value on memory formation, including the present study, presented the information to be remembered (i.e. the memoranda) in close temporal vicinity to the feedback^{9,14,18,20}, while those reporting no or a negative impact of feedback value presented the memoranda prior to the feedback^{15,21}. Interestingly, Jang et al. (2019) used a paradigm where rewards were presented either before memoranda (to induce reward anticipation), or during memoranda (to elicit, what the authors termed, an “image prediction error” associated with the image category), or after memoranda (to elicit a feedback prediction error). Strikingly, while neither the feedback prediction error nor the value of reward anticipation impacted on subsequent memory performance, the image prediction error, elicited by the presentation of memoranda, correlated positively with subsequent memory performance. Thus, there might be a narrow time-window for reward delivery to enhance memory encoding, as further indicated by the very rapid phasic response of dopamine neurons to reward delivery (<500ms; ²²). Yet, because a few other studies showed that activating the reward system before and after memoranda also increased subsequent memory performance^{7,23}, further research is clearly needed to determine to what extent memory formation depends on the relative timing between memoranda and different aspects of reward, and how these relate to the different response profiles of the dopamine system, i.e. phasic bursts and dips, tonic activity, and sustained ramping responses^{9,24}.

3. I was a little unclear about the randomization of average reward values across characters. Was this factor randomized across individual participants, or where the same reward values used for a given character for each participant. If there was no randomization, there could be confounds of character identity. Also did the authors account for any differences in interference of accumulating reward information across a given identity, or testing-order effects across identities. Supplemental analyses on these important factors should be including.

Thank you for raising these highly relevant points. As a reply to your first concern, the characters were randomly assigned to an average reward category for each participant. This critical information has now been added to lines 143-145, which now reads:

For each participant, the characters were randomly assigned to one of the different average reward levels (i.e. either Lo, Me, or Hi).

To address your concern regarding potential confounding influences from character identity and/or presentation order, we controlled for i) trial number during testing, ii) trial number during encoding, and iii) character identity, by including these three factors in a linear-mixed effects model together with the factors of interest Feedback value and Average reward. Participant identity was included as a random factor. Any significant effects of Feedback value and Average reward in this new model can thus be interpreted as being independent from trial number and character identity. The main effects of Feedback value [$F(2, 1945) = 22.28, p < 0.001$, ANOVA] and Average reward [$F(2, 1945) = 3.55, p = 0.029$, ANOVA] remained significant after controlling for these potentially confounding parameters.

This analysis was added to the Supplementary Information (Supplementary Note 1, and is now mentioned in the main text in lines 339-342:

The main effects of Feedback value and Average reward remained significant when controlling for presentation order during encoding, presentation order during testing, and cartoon character identity (see Supplementary Note 1).

To further address your concern in relation to "...differences in interference of accumulating reward information across a given identity ...", we tested the fits of two new computational models that extend the most parsimonious model of the first version of the manuscript. The first of these new models allows for inter-character differences in initial levels of average reward (i.e. for each participant six different initial levels of average reward were fitted), while a second model allows different rates of accumulating average reward (i.e. for each participant six different learning rates were fitted). In brief, compared to the most parsimonious model of the first version of the manuscript [mean AIC=107.67], both of these new models provided inferior fits to behavior [mean AIC different initial values=115.07, $t(32)=4.44$, $p<0.001$, paired t -test; mean AIC different learning rates=113.85, $t(32)=4.58$, $p<0.001$, paired t -test].

This analysis was added to the Supplementary Information (Supplementary Note 2 and the results are acknowledged in lines 350-352 of the main text:

The $R_{fb+\bar{r}^2}$ model also provided the most parsimonious fit as compared to models fitting separate initial average reward levels and learning rates for each character (see Supplementary Note 2).

4. Modeling is not my specific expertise, but I was unclear whether average reward values was modeled for each character separately or if it was run as a continuous value across the experiment. I believe it was the latter. If so, I think the authors should also test a model in which each character identity starts with a given baseline and is updated separately. Specifically, participants could be forming higher-order representations (i.e., latent causes) in which average reward value is segmented by character identity. If so, this could have consistent, albeit subtle, differences on the relationship between average reward and subsequent memory/neural activation.

Thank you for raising this important point. You are correct in that the level of average reward was calculated independently of character identity. Following your suggestion, we also tested whether average reward was calculated for each character independently by fitting a model in which the average reward level was set to a fitted value ($C_{\bar{r}_0}$) the first time a new character was presented. Besides the addition of the $C_{\bar{r}_0}$ parameter, the other parameters of this new model were the same as for the most parsimonious model from the first version of the manuscript. This new model provided an inferior fit as compared to the most parsimonious model of the first manuscript [mean AIC most parsimonious model=107.67; mean AIC new model=111.12, $t(32)=3.06$, $p=0.004$, paired t -test]. In other words, the impact of average reward on memory encoding is better described via a "global" calculation of average reward, rather than a "local" calculation that is constrained to a particular character identity.

The fitted parameters of this model is now included in Table 1 and we added a description of this model to the Methods section (see line 220-222):

A seventh and final model is similar to the sixth model, but presumes that average reward levels are calculated for each cartoon character independently. To account for this possibility, an initial level of average reward ($C_{\bar{r}_0}$) was therefore fitted across cartoon characters.

Please observe that fitting six different C_{T0} parameters (i.e. one for each cartoon character), did also not improve the model fit (see our response in the previous paragraph).

5. An 8 mm smoothing kernel is quite large for the VTA, and I am worried that the authors could also be incorporating signal from surrounding midbrain structures (i.e., SN, red nucleus). Do the results remain the same using a smaller kernel (i.e., 4 mm) or a more ROI based approach without smoothing.

Thank you for raising this issue, which is highly relevant given the supposedly pivotal role of the VTA in enabling reward-related memory enhancements. Based on your suggestions, we performed additional ROI-based analyses by extracting and averaging beta-values across all voxels within a VTA ROI. To confirm the robustness of our results, we tested two different smoothing kernels (the 8mm kernel reported in the first version of the manuscript and the 4mm kernel as per your suggestion) and two different VTA ROIs. The first VTA ROI (used in the first version of the manuscript) was based on a probabilistic atlas of the VTA restricted to include only voxels shared by at least 50% of the participants²⁵. The second VTA ROI was based on the coordinates of a previous study of reward-related memory enhancements²³. In brief, this second VTA ROI was created by centering two 4mm radius spheres on the Talairach coordinates [-4 -15 -9; 5 -14 -8] transformed to MNI space. We previously used this ROI to study the neural correlates of prediction errors in the dopaminergic midbrain²⁶.

These analyses revealed that VTA BOLD signal was non-linearly modulated by average reward for all four combinations of ROIs and smoothing kernels (all p-values < 0.05), thus confirming the robustness of our initial results.

We now provide a detailed explanation of these analyses and the results in the Supplementary information (Supplementary Note 5). In the main text, this result is now acknowledged in lines 442-444 as:

Additional supplementary analyses, using a more conservative ROI approach, confirmed that the effect of average reward on VTA activity was robust across different VTA ROIs and smoothing kernels (see Supplementary Note 5).

6. In regards to the bar graphs reporting the functional activations, I think rather than illustrating the peak the presentation of results should be extracted from the entire cluster. Further, statistics should be run on these peak extracted clusters as this results in a circular analysis. Rather the plots should just be used for visualization purposes (see Kriegeskorte et al., 2010 for discussion).

Regarding the issue of circularity, we sincerely thank you for pointing out this oversight on our behalf. The sole purpose of showing the extracted beta parameters was to visualize the effects of the parametric modulators, because these are difficult to interpret otherwise (in particular the non-linear modulation by average reward). To correct this oversight, we have now removed all asterisks and notations related to statistical significance testing in the Figures. Moreover, we now explicitly mention that the extracted beta parameters are merely for display purposes in the Methods section as well as in every Figure legend.

Specifically, in lines 307-308 of the Methods:

For display purposes, the average brain activity were extracted from 3mm spheres centered on peak voxels within significant clusters of activation and shown as insets.

Example of a line added to the Figure legends:

“For display purposes, the violin plots show the average % signal change extracted from 3mm spheres centered on peak voxel coordinates for each significant activation cluster.”

However, we disagree that it is more suitable to plot the average data within an activated cluster rather than the peak voxel activity. We believe it is more appropriate to display the data which the statistics is based on (i.e. here, the peak activation). Additionally, extracting the average data from all voxels within an activated cluster raises issues regarding activation that extends beyond anatomical boundaries of a pre-defined ROI. Finally, although extracting the average data within a cluster of activation is one of the preferred methods when performing subsequent between-subject analyses, such as inter-individual correlations between BOLD signal and measures of task performance or personality traits ²⁷, we know of no such recommendations for visualizing group-level data.

Yet, we now partially acknowledge the reviewer's request by visualizing the average beta-parameters extracted from 3mm radius spheres centered on the peak-voxel activation (i.e. rather than just visualizing the activity of the peak voxel). We would certainly consider plotting the average activation within an activated cluster (bounded by a priori anatomical ROIs) instead, if requested by reviewer.

All Figures displaying extracted beta parameter estimates have now been updated with the new information (i.e. Figures 3, 4, and 5).

7. More details on the anatomical specificity of the hippocampal activation would be helpful. It is overlapping across analyses? Is it in more anterior or posterior portions. In a few samples it seems like the peaks are in cortical MTL rather than hippocampus. Perhaps presenting the data on a sagittal view (either in the manuscript or supplementals) would provide a better representation of the anatomical locations.

Thank you for these relevant suggestions. To better visualize activation within the temporal lobe, we now provide sagittal views in the Supplementary information (Supplementary Note 6) that also include the ROIs used for the data analyses (i.e. the hippocampus, the parahippocampal gyrus, and the amygdala). For all relevant analyses, we acknowledge these additional views by the sentence:

Sagittal views of these activations are provided in Supplementary Note 6.

We believe these additional views can now be used to resolve your question regarding activation along the anterior-posterior portions of the hippocampus.

To address your question regarding overlapping activation across analyses, we conducted a conjunction analysis for the contrasts of the parametric modulators Feedback value and Average reward (i.e. the inverted U-shape). Specifically, we compared the conjunction of these contrasts to the conjunction null hypothesis (i.e. testing for voxels significantly activated by both contrasts; the equivalence to the logical AND ²⁸). Using an uncorrected threshold of $p=0.001$, no voxels within our a priori ROIs were significantly activated in both contrasts.

We now acknowledge this result in the main text in lines 446-449:

Moreover, the conjunction between the BOLD signal modulated by average reward (inverted U-shape) and Feedback value revealed that no voxels in the a priori ROIs were significantly modulated by both Feedback value and Average reward (the conjunction was tested versus the ‘conjunction null hypothesis’²⁸).

Reviewer #2 (Remarks to the Author):

In this study, authors used computational modeling and fMRI to test the behavioral and neural modulations by immediate and average reward amount during memory encoding, and the relationship between the two types of modulation. The manuscript is very clearly written and clever experimental manipulations were adopted to distinguish encoding modulation by immediate and average reward amount. Dissociations between so-called phasic and tonic DA activities and their distinct contribution to memory encoding is not terribly new, but showing them in one simple empirical paradigm, along with the interaction of DA system with dACC, makes a new contribution to the literature. I have a few concerns and questions that I feel needed to be addressed before this work is published.

1. I have a few questions about the way fMRI data was modeled and the rationale behind it. If I understood correctly, the onset of stimuli and feedback display were modeled in each trial, and two parametric regressors (model-derived estimates of immediate reward and average reward levels) were added to ‘feedback onset’.

1a. Most simply, how were the events modeled? Was a stick function or a boxcar function was used? The term “event-related design” in the fMRI analysis description implies that a stick function was used, but it needs to be explicitly described.

Thank you for noticing this omission. The events were indeed modeled as stick functions. We added this information to the manuscript by updating the fMRI model-description accordingly (see lines 262-264):

To this end, we created an event-related design that included three event-types **modeled as stick functions (i.e. with a 0 duration)** that were respectively time-locked to the onset of the display of character/object pairs, **the button press of the response**, and the feedback in each trial.

1b. Why was ‘response’ not modeled? This would leave response-related variations of the BOLD signal unaccounted for, possibly adding noise to the results. I understand that response was not an event of interest for any analyses in this study, but it still needs to be included in the model.

Thank you for this relevant suggestion. We have now added a new ‘response’ regressor at the onset time of the response. This addition is now acknowledged in line 262-264:

To this end, we created an event-related design that included three event-types **modeled as stick functions (i.e. with a 0 duration)** that were respectively time-locked to the onset of the display of character/object pairs, **the button press of the response**, and the feedback in each trial.

We have updated all the Tables, Figures, and the text with the results obtained from the model including the response regressor.

Importantly, adding the response regressor to the model caused only slight changes in the numerical values of the results, i.e. effects reported to be significant without the response regressor remained significant also in the new model that included the response regressor.

1c. This last point is of more importance and applies broadly to other parts of the manuscript (including Introduction and interpretations). In essence, my point is about

the assumption and approaches about when/how immediate and average reward amount would influence memory encoding. Encoding of a specific event starts with the onset of the event (or, stimulus display in the experiment) and the continuing encoding process gets influenced by feedbacks. Critically, where the action occurs in this line of process (starting from the stimulus display and continuing till feedback, or the end of the event) may differ for immediate and average reward. For example, whereas immediate reward feedback is likely to affect encoding of the event that just happened, average reward level is more likely to affect encoding of the upcoming event.

In the present study, the GLM included parametric regressors at the onset of feedback display, which will get at “whether there is any additional activation modulated as a function of the parametric regressors (i.e., linear effects of immediate reward and inverted-U shape effects of avg reward) when the average activation during feedback display is accounted for”. This makes sense for the effects of immediate reward on DA system and memory encoding, however, when it comes to the effects of average reward level, there would be at least 2 different processes to consider. Firstly, when information comes in (stimuli presentation), the motivational state, or tonic/sustained DA activation is likely to impact encoding of this information. Then at the time of feedback onset, the representation of average reward level gets adjusted – by integrating the running reward average up to that moment and the reward amount in the feedback that was just received. Given the way data were modeled, the present results likely reflect the latter, and it makes total sense that the VTA activation during this feedback integration phase is tightly linked with dACC activation. This in itself is a meaningful finding, but it does not address exactly what the study is proposing to test. When it comes to the effects of average reward level on memory encoding, the action is most likely to be happening at the onset of stimuli display.

Thank you for this highly relevant insight. Initially, we did not report any reward-related modulation of BOLD signal at the character-objects presentation (i.e. stimulus presentation) because we hypothesized that the associative memory formation would be modulated by the effect of reward on feedback only. This assumption was based on the fact that the critical information to be encoded (i.e. whether a character liked or disliked a selected object) was only revealed during the feedback presentation.

However, we certainly agree with your suggestion that the level of average reward may also impact on the encoding of individual stimuli. On that note, although the present task was not designed to test memory for individual stimuli, it could easily be re-designed to test the impact of average reward on memory also for individual stimuli, for example by replacing the test for character-object associations with a recognition memory test.

To highlight this issue, we added one paragraph to a new Limitations section of the Discussion which reads (lines 664-669):

The present study shows that average reward modulates the encoding of character-object associations, yet it remains unknown to what extent average reward modulates the encoding of other types of information within a trial, such as memory for a particular character identity or objects. While the present study was not designed to test memory for individual stimuli, the experimental paradigm could easily be adapted to test such aspects by, for example, replacing the associative memory test with a stimulus recognition test.

To address your specific concern, i.e. whether neuronal activity at stimulus presentation was also modulated by average reward, we tested an additional fMRI model in which only the BOLD signal at stimulus presentation was parametrically modulated. Identical to the main fMRI model of the manuscript, this new model contained three event-types (Stimulus presentation, Button press at response, and Feedback) modeled as stick-functions. However, the parametric modulators Feedback value and Average reward were now added only to the Stimulus presentation (and not to the Feedback as in the main model). In brief, this new model revealed no significant modulation of the BOLD signal at stimulus presentation by Feedback value or Average reward within the a priori ROIs. While the null-effect of Feedback value was expected (because the feedback had yet to be presented), the lack of modulation by Average reward supports that average reward exerts its effect at the time of feedback processing only, namely at the time when the relevant information to be encoded in memory (object-character association) is revealed.

We now provide a detailed explanation of this additional fMRI analysis and the results in the Supplementary Information (Supplementary Note 7). Because no voxels in our a priori ROIs were significantly activated even at an uncorrected threshold of $p=0.001$, we opted to visualize these null-results using glass brains. Specifically, for the parametric modulation of Feedback value and Average reward (inverted U-shape) at stimulus presentation, we now show all brain activations at an uncorrected threshold of $p=0.001$, as well as brain activations at an uncorrected threshold of $p=0.05$ inclusively masked by our a priori ROIs:

Supplementary Figure 4. Glass brains showing BOLD signal at the onset of character-object pairs. A. Parametric modulation by Feedback value at uncorrected threshold $p=0.001$, no inclusive mask. B. Parametric modulation by Feedback value at uncorrected threshold $p=0.05$, inclusively masked by the hippocampus, parahippocampal gyrus, amygdala, ventral tegmental area, and the nucleus accumbens. C. Parametric modulation by Average reward (inverted U-shape) at uncorrected threshold $p=0.001$, no inclusive mask. B. Parametric modulation by Average reward (inverted U-shape) at uncorrected threshold $p=0.05$, inclusively masked by the hippocampus, parahippocampal gyrus, amygdala, ventral tegmental area, and the nucleus accumbens.

Finally, this result is acknowledge in lines 444-446 of the main text:

Of note, a separate fMRI analysis confirmed that average reward had no impact on the BOLD signal evoked during stimulus presentation (see Supplementary Note 7).

2. The point 1c above applies to modeling of behavioral data as well. It was surprising to see that the influence of immediate vs. average reward level on memory encoding was independent of each other. In fact, again, I suspect that it may be because of the way data were modeled. Encoding of a given trial would most influenced by the immediate reward of that trial and the running average reward up to that stimulus display (rather than average reward update following the feedback). It would make more sense to test whether the average reward level activated by the onset of stimulus display (a character that is assigned to Lo, Me, or Hi average reward level) modulates the effects of immediate reward amount to memory encoding (interaction).

If this alternative model also reveals independent influence of the two on encoding, it would be important to discuss it in depth.

Thank you for these relevant comments, which we will address in detail below.

We would like to start by re-iterating some results that are directly related to your concern regarding the lack of interaction between immediate and average reward:

- The behavioral results also showed a non-significant interaction between immediate reward and the average reward level (the latter being estimated by the Lo, Me, and Hi characters; Feedback value x Average reward interaction: $F(4,288)=1.008$, $p=0.404$, ANOVA). The most parsimonious model thus correctly replicates this non-significant behavioral result [model-predicted interaction: $F(4,288)=0.850$, $p=0.495$, ANOVA].
- In one of the tested models we included an interaction term between immediate reward and average reward (see model five in the main text):
$$R_{fb+\bar{r}+fb\bar{r}}(t) = C_0 + C_{fb} * fb(t) + C_{\bar{r}} * \bar{r}(t) + C_{fb\bar{r}} * fb(t) * \bar{r}(t).$$
This model provided an inferior fit to behavior, as compared to the most parsimonious model [mean AIC most parsimonious model=107.67; mean AIC interaction model=114.23, $t(32)=4.86$, $p<0.001$, paired t -test].

These results therefore support the notion of independent influences from immediate and average reward on memory formation.

Regarding your concern related to “**Encoding of a given trial would most influenced by the immediate reward of that trial and the running average reward up to that stimulus display (rather than average reward update following the feedback)**”, we want to clarify that the most parsimonious model actually does model the impact of average reward on memory encoding as the average reward up to that stimulus display, and not the average reward update. This model is therefore in accordance with your suggestion.

Regarding your final concern, i.e. “**It would make more sense to test whether the average reward level activated by the onset of stimulus display (a character that is assigned to Lo, Me, or Hi average reward level) modulates the effects of immediate reward amount to memory encoding (interaction).**”, we reported in the previous paragraph a complementary fMRI analysis showing that BOLD signal at stimulus presentation was not modulated by average reward. Thus, it is unlikely that the effect of average reward on the encoding of character-object associations is related to processes evoked by the stimulus presentation.

However, your suggestions do highlight the possibility that the reported memory-modulations may be caused by reward anticipation evoked by the presentation of a particular cartoon character, a possibility we did not consider up to now. In other words, the calculation of average reward may have been constrained to a particular character and would thus be reset whenever another cartoon character was presented. This “local” calculation of average reward contrasts with the more “global” calculation of average reward used in the first version of the manuscript.

We addressed this question via a computational approach. Specifically, to test whether “local” calculations of average reward provided a better fit to behavioral data, we designed two new models that constrained the calculation of average reward levels to specific characters. One model fitted one baseline level of average reward across all characters, and this baseline level was used as the initial level of average reward for each character. A second model fitted one baseline level for each character separately (i.e. six different baselines were fitted for each participant). In brief, both of these models provided inferior fits to the most parsimonious model of the previous version of the manuscript [mean AIC parsimonious model=107.67; mean AIC one baseline model=111.12; mean AIC six baselines model=115.07; all p-values < 0.05, paired *t*-tests].

Thus, the impact of average reward on memory encoding is better described by a “global” rather than a “local” calculation of average reward.

The description of one of these models was added to the main text at lines 220-222, with the fitted parameters shown in Table 1:

A seventh and final model is similar to the sixth model, but presumes that average reward levels are calculated for each cartoon character independently. To account for this possibility, an initial level of average reward ($C_{\bar{r}0}$) was therefore fitted across cartoon characters.

The description and the results of the second model was added to the Supplementary Information (Supplementary Note 2).

We also acknowledge that average reward is more likely calculated on a “global” level, rather than “locally” in lines 376-381:

Of note, the level of average reward was not constrained to particular character identities, as the most parsimonious model provided a better fit as compared to other models fitting i) one baseline value of average reward which was reset whenever a new character was presented during encoding (see the $R_{fb+\bar{r}^2}$ with $C_{\bar{r}0}$ model), or ii) six different baseline levels of average reward (i.e. one for each cartoon character; see Supplementary Note 2).

3. Very closely related to the point above (1c & 2) in the Introduction, authors bring up two possible scenarios (not necessarily against each other) on the sustained effects of reward on memory encoding – interactions between tonic and phasic dopaminergic activities, and top-down control of VTA by dACC. Results clearly support the latter, but not much is discussed as to what the results mean with regards to the tonic-phasic DA interaction story. Does the lack of interaction effects on memory encoding rule out this possibility? Applying the a parametric regressor of average reward level to the onset of stimulus display (the average before being updated with the current trial reward amount) may reveal meaningful interactions between tonic and phasic DA interactions.

Thank you for these suggestions. Indeed, we did not thoroughly elaborate on the meaning of our results in the light of phasic and tonic dopamine, in particular regarding the relationship between tonic dopamine and the dACC-VTA interaction. Actually, one plausible option is that the top-down modulation by the dACC leads to alterations in VTA tonic dopamine activity. Thus, we now present a more elaborate discussion on how the dACC may modulate tonic dopamine activity in the VTA (see lines 570-580):

While the neuronal mechanisms in the VTA that are impacted by a dACC-VTA modulation are largely unknown, the ACC projects directly to VTA dopamine neurons via glutamatergic projections^{29, 30, 31, 32}, and glutamatergic afferents to dopamine neurons supposedly control transitions between phasic and tonic activity^{33, 34, 35}. Moreover, shifting the balance between excitatory glutamatergic and inhibitory GABAergic inputs to the VTA increased the spontaneous activation of dopamine neurons³⁶, and glutamate iontophoresis increased both the baseline firing rate and the burst firing frequency of dopamine neurons³⁷. Thus, one plausible mechanism, which certainly needs empirical confirmation, is that top-down control of the VTA from the dACC occurs via glutamatergic inputs to the VTA that alter tonic dopamine activity by changing the number of spontaneously 'active' dopamine neurons (i.e. neurons that may elicit phasic burst firing), and thus also the overall phasic dopamine response^{34, 38, 39}.

4. Ventral striatum is a main target of VTA DA projection, and it is surprising to see completely different reward-related response profiles in vSTR and VTA. Results suggest that VTA is NOT sensitive to immediate reward amount, and instead its activation changes in an inverted-u shaped function of average reward update - with the greatest activation increases at medium level average reward condition. vSTR, which gets downstream signal from VTA, however, is NOT sensitive to the average reward level, but instead exhibits greater activation in response to greater immediate feedback. I believe this is worth a discussion point - what are authors' thoughts on this?

Thank you for your comment, which prompted us to further corroborate this finding and improve its interpretation. To test whether the lack of activation in the VTA by immediate reward may have been caused by the selection of a rather conservatively defined VTA ROI, we tested whether beta parameter estimates for all voxels within a functionally defined VTA ROI were significantly modulated by Feedback value. This VTA ROI was based on coordinates obtained from a previous study of reward-related memory enhancement²³, which we previously used to test the neural correlates of prediction errors in the dopaminergic midbrain²⁶. In brief, this VTA ROI was created by centering two 4mm radius spheres on the Talairach coordinates [-4 -15 -9; 5 -14 -8] transformed to MNI space. This analysis revealed that on average the voxels in this functionally defined VTA ROI tracked Feedback values. Using a small-volume correction procedure (with a search threshold of $p=0.001$, uncorrected) also showed that voxels within this VTA ROI was significantly activated by Feedback value [MNI: 6 -10 -8, $T(32)=3.37$, $p_{SVC, FWER}=0.011$].

The conservative ROI analysis was added to the Supplementary Information (Supplementary Note 4), and is now acknowledged in the main text (lines 403-412):

However, given the well-known role of the VTA in reward processing and reward-related memory enhancements, it was surprising that no voxels with the VTA ROI tracked feedback value. To test whether this null-result may have been related to the selection of an anatomically defined VTA ROI, in combination with a strict requirement for the number of overlapping voxels across participants (50%), we performed an additional analysis using a conservative ROI approach in a

functionally defined VTA ROI. This VTA ROI was based on coordinates obtained from a previous study looking at reward-related memory enhancement²³, and we previously used this VTA ROI to test prediction error encoding in the dopaminergic midbrain²⁶. This supplementary analysis revealed that BOLD signal in this VTA ROI indeed correlated with feedback values (see Supplementary Note 4).

Additionally, we also updated Table 2 accordingly:

Table 2. BOLD signal showing positive correlation with increasing feedback values in a priori ROIs and the amygdala ROI. $p_{FWE, SVC}$ indicates the p-value resulting from familywise error (FWE) small volume correction (SVC) on peak voxel activity within a priori ROIs and the amygdala ROI.

	Hemisphere	MNI peak coordinate			T(32)	$P_{FWE, SVC}$
		x	y	z		
Positive correlation with feedback value						
Reward mask:						
Nucleus accumbens	Left	-18	5	-14	9.085	<0.001
Nucleus accumbens	Right	15	8	-11	7.593	<0.001
VTA ¹						
Memory mask:						
Hippocampus / Parahippocampal gyrus	Left	-15	2	-20	6.724	<0.001
		-24	-13	-14	4.310	0.031
		-21	-7	-20		0.032
Hippocampus / Parahippocampal gyrus	Left	-21	-28	-8	4.329	0.030
Hippocampus / Parahippocampal gyrus	Right	27	-34	-5	5.004	0.006
		18	5	-20	4.522	0.019
Amygdala ROI (post-hoc) :						
Amygdala ²	Left	-18	2	-23	6.147	<0.001
Amygdala ²	Right	18	2	20	4.069	0.017 ns

¹No voxel within the anatomically defined VTA ROI included in the “reward mask” was significantly activated by feedback value. However, a complimentary ROI analysis using a slightly different and functionally defined VTA ROI showed that VTA BOLD signal significantly tracked feedback value (see supplementary information for details). ²The amygdala was not part of the initial hypotheses, thus a stricter (Bonferroni-corrected) statistical threshold was applied in order to infer any involvement of the amygdala ($\alpha=0.0167$). ns=not significant.

To elaborate upon the finding that the NAcc tracked feedback value, but not average reward, we added a new section to the Discussion, which reads:

Divergent encoding of feedback value and average reward between the NAcc and the VTA

The VTA encoded both feedback value and average reward (inverted U-shape) while the NAcc only encoded feedback value. At first glance, the divergent encoding of average reward in the VTA and the NAcc is surprising given the extensive dopaminergic input from the VTA to the NAcc⁴⁰, in combination with results showing co-activation of the VTA and the NAcc in the context of

prediction errors^{41, 42, 43} and during rest^{25, 44}. However, the NAcc receives input from many brain regions besides the VTA⁴⁵, and may therefore be influenced by information beyond that transmitted by VTA dopamine neurons⁴⁶. In line with this notion, one study alternated visual and auditory cues as being the relevant feature to predict upcoming rewards, and showed that the ventral striatum (VStr; of which the NAcc is part) tracked the value of the currently relevant stimulus only, and was activated by the correctness of the response rather than the outcome's value⁴⁷. In a related study, participants needed to learn the timing of a reward outcome rather than its magnitude, and it was reported that while the VTA tracked parameters related to both the magnitude and the timing of reward, the VStr only tracked the task-relevant parameters, i.e. the ones related to the outcome timing. Similar to this latter study, participants in the present study needed to learn the value of character-object associations, and this task-relevant feature was tracked by both the VTA and the NAcc. By contrast, the average reward level was irrelevant to the task and was tracked only by the VTA. These results support the notion that the ventral striatum/NAcc updates the features of the environment which are most relevant to drive behavior⁴⁸, a specific role that seems not to be shared by the upstream VTA.

5. A proper group-level inference of the model fit for behavioral results would help succinct presentation of findings. Table 1 shows an example participant's data & Figure 2E shows the model estimated values – which resembles the pattern in the actual data (Figure 1F, G). A more succinct and convincing way to draw a group-level inference would be to use a mixed-model to fit the data from all subjects in one model while controlling for variability from subjects.

Thank you for bringing up this important point. To confirm that the model-predicted behavior resembles actual behavior also statistically, we used the fitted parameters of each participant to calculate a model-predicted memory encoding probability in each trial. Then, as per your suggestion, we performed a linear mixed-effects model with Feedback value (-1, +1, +5) and Average reward (Lo, Me, Hi) as fixed effects, and participant as random effect. As with actual behavior, the model-predicted behavior revealed significant main effects of Feedback value [F(2, 288)=61.640, p<0.001, ANOVA] and Average reward F(2,288)=11.636, p<0.001 ANOVA], but no significant interaction between them [F(4,288)=0.850, p=0.495, ANOVA]. Pairwise comparisons further confirmed that the model predicted highest memory performance for medium (Me) average reward, as compared to both Lo and Hi average reward (all p-values<0.001, paired *t*-tests), as well as better memory performance for higher feedback values (i.e. +5>+1>-1; all p-values<0.001, paired *t*-tests).

We added these supplementary analyses to the Supplementary Information (Supplementary Note 3), and refer to these results in the main text in lines 375-376:

The fit of the $R_{fb+C_{rr}}^2$ model to behavioral data is displayed in Figures 2E and 2F, and shows the same significant effects as actual behavior (Figure 1F,G; **see Supplementary Note 3**).

Moreover, for consistency we replaced the repeated measures ANOVA used to analyze the behavioral results with a linear mixed-effects model. As would be expected, all significant and non-significant effects of the repeated measures ANOVA remained respectively significant and non-significant also for the linear mixed-effect model [Feedback value: F(2, 288)=17.905, p<0.001, ANOVA; Average reward: F(2,288)=4.792, p=0.009 ANOVA; Feedback value x Average reward interaction: F(4,288)=1.008, p=0.404, ANOVA].

The main text has been updated accordingly, and we also now describe the linear mixed-model in the Methods section, lines 173-189:

The data were analyzed using linear mixed models of the following form: $Y_i = X_i B + Z_i b_i + e_i$, where Y_i represents a vector of values of the dependent measure of interest for the i th participant, X_i represents a matrix of p predictors (independent variables) for the i th participant, B represents a vector of p fixed effect beta weight estimates for each predictor in X_i , Z_i represents a matrix of q random effect predictors, b_i represents a vector of q random effect estimates, and e_i represents a vector of the model fit error, corresponding to the discrepancy between the model prediction for each observation from the i th participant and the actual value of that observation. For behavior and modeling data, two categorical predictors with three levels each were used: Feedback value (-1, +1, +5) and Average reward (Lo, Me, and Hi) along with their interaction. Moreover, each participant was treated as a random variable, i.e. the matrix Z contained one column for each participant pertaining to that participant's random effect estimate b .

ANOVA was initially used to analyze the results, and significant effects were further investigated using paired t -tests. Of note, while linear mixed models are in many ways superior compared to alternative approaches^{49,50}, the calculation of standard effect sizes for these models is still heavily debated⁵¹. For this reason, we restricted the reporting of standardized effects sizes to the difference between means, which was calculated using Cohen's d :

$$d = \frac{\mu_1 - \mu_2}{\sqrt{(\sigma_1^2 + \sigma_2^2)/2}}$$

6. It was a bit unclear how ROIs were used in analyses with the two different parametric regressors. Were both the linear and non-linear parametric regressors effects tested in all ROIs? Or were particular sets of ROIs were tested for each parametric regressor?

Thank you, it was indeed not well described which ROI was tested for which analysis. We tested whether "reward-related" and "memory-related" brain regions were activated by Feedback value and Average reward. To this end, we created a "reward-mask" consisting of a NAcc ROI and a VTA ROI, and one "memory-mask" consisting of a HC ROI and a PHG ROI. By contrast, the dACC mask was only used to test the specific hypothesis regarding the dACC-VTA interaction. Additionally, the amygdala ROI was added post-hoc and was therefore subjected to appropriate Bonferroni-corrections. We now provide a better description of the ROIs and the tests they were part of in lines 290-294:

A "reward mask" was created by combining the NAcc and VTA ROIs, while a "memory mask" was created by combining the HC and the PHG ROIs. Both of these masks were used to test the parametric modulation by Feedback value and Average reward. The dACC ROI was only used to test specific hypotheses related to VTA-dACC interactions.

We also updated Tables 2 and 3 to acknowledge the different masks, for example Table 2 now reads:

Table 2. BOLD signal showing positive correlation with increasing feedback values in a priori ROIs and the amygdala ROI. $p_{FWE, SVC}$ indicates the p-value resulting from familywise error (FWE) small volume correction (SVC) on peak voxel activity within a priori ROIs and the amygdala ROI.

	Hemisphere	MNI peak coordinate			T(32)	$P_{FWE, SVC}$
		x	y	z		
Positive correlation with feedback value						
Reward mask:						
Nucleus accumbens	Left	-18	5	-14	9.085	<0.001
Nucleus accumbens	Right	15	8	-11	7.593	<0.001
VTA ¹						
Memory mask:						
Hippocampus / Parahippocampal gyrus	Left	-15	2	-20	6.724	<0.001
		-24	-13	-14	4.310	0.031
		-21	-7	-20		0.032
Hippocampus / Parahippocampal gyrus	Left	-21	-28	-8	4.329	0.030
Hippocampus / Parahippocampal gyrus	Right	27	-34	-5	5.004	0.006
		18	5	-20	4.522	0.019
Amygdala ROI (post-hoc) :						
Amygdala ²	Left	-18	2	-23	6.147	<0.001
Amygdala ²	Right	18	2	20	4.069	0.017 ns

¹ No voxels within the anatomically defined VTA ROI included in the “reward mask” were significantly activated by feedback value. However, a complimentary ROI analysis using a slightly different and functionally defined VTA ROI showed that VTA BOLD signal significantly tracked feedback value (see supplementary information for details). ² The amygdala was not part of the initial hypotheses, thus a stricter (Bonferroni-corrected) statistical threshold was applied in order to infer any involvement of the amygdala ($\alpha=0.0167$). ns=not significant.

7. There is an inconsistency in terminology for NAcc, or ventral striatum. NAcc is not mentioned until Discussion – My guess is that authors are referring to the vSTR as NAcc there (or the other way around). NAcc is included in vSTR, but they are not the same, therefore should not be referred to interchangeably. Which one was the actual ROI?

Thank you for spotting this inconsistency. Because we used a NAcc ROI, we have updated the text accordingly by replacing any reference to the VStr (in the context of our results) as the NAcc.

8. Typos [Line 127]: “(three +1’s, one +5’s)” – Isn’t this supposed be “(three +1’s, three +5’s)”?

Thank you for spotting this mistake on our behalf. We have updated the Figure legend accordingly.

References

1. Lisman J, Grace AA, Duzel E. A neoHebbian framework for episodic memory; role of dopamine-dependent late LTP. *Trends in neurosciences* **34**, 536-547 (2011).
2. Bliss TV, Collingridge GL, Morris RG, Reymann KG. Long-term potentiation in the hippocampus: discovery, mechanisms and function. *Neuroforum* **24**, A103-A120 (2018).
3. Lisman JE, Grace AA. The hippocampal-VTA loop: controlling the entry of information into long-term memory. *Neuron* **46**, 703-713 (2005).
4. Wang SH, Morris RGM. Hippocampal-Neocortical Interactions in Memory Formation, Consolidation, and Reconsolidation. *Annual Review of Psychology* **61**, 49-79 (2010).
5. Bethus I, Tse D, Morris RGM. Dopamine and Memory: Modulation of the Persistence of Memory for Novel Hippocampal NMDA Receptor-Dependent Paired Associates. *Journal of Neuroscience* **30**, 1610-1618 (2010).
6. O'Carroll CM, Martin SJ, Sandin J, Frenguelli B, Morris RGM. Dopaminergic modulation of the persistence of one-trial hippocampus-dependent memory. *Learning & memory* **13**, 760-769 (2006).
7. Murayama K, Kitagami S. Consolidation power of extrinsic rewards: reward cues enhance long-term memory for irrelevant past events. *Journal of experimental psychology General* **143**, 15-20 (2014).
8. Wittmann BC, Schott BH, Guderian S, Frey JU, Heinze HJ, Duzel E. Reward-related FMRI activation of dopaminergic midbrain is associated with enhanced hippocampus-dependent long-term memory formation. *Neuron* **45**, 459-467 (2005).
9. Stanek JK, Dickerson KC, Chiew KS, Clement NJ, Adcock RA. Expected Reward Value and Reward Uncertainty Have Temporally Dissociable Effects on Memory Formation. *Journal of cognitive neuroscience* **31**, 1443-1454 (2019).
10. Patil A, Murty VP, Dunsmoor JE, Phelps EA, Davachi L. Reward retroactively enhances memory consolidation for related items. *Learning & memory* **24**, 65-69 (2017).
11. Igloi K, Gaggioni G, Sterpenich V, Schwartz S. A nap to recap or how reward regulates hippocampal-prefrontal memory networks during daytime sleep in humans. *eLife* **4**, (2015).
12. Perogamvros L, Schwartz S. The roles of the reward system in sleep and dreaming. *Neuroscience and biobehavioral reviews* **36**, 1934-1951 (2012).

13. Rasch B, Born J. About Sleep's Role in Memory. *Physiological reviews* **93**, 681-766 (2013).
14. Aberg KC, Muller J, Schwartz S. Trial-by-Trial Modulation of Associative Memory Formation by Reward Prediction Error and Reward Anticipation as Revealed by a Biologically Plausible Computational Model. *Frontiers in human neuroscience* **11**, 56 (2017).
15. Jang AI, Nassar MR, Dillon DG, Frank MJ. Positive reward prediction errors during decision-making strengthen memory encoding. *Nature human behaviour* **3**, 719-732 (2019).
16. Bialleck KA, Schaal HP, Kranz TA, Fell J, Elger CE, Axmacher N. Ventromedial prefrontal cortex activation is associated with memory formation for predictable rewards. *PLoS one* **6**, e16695 (2011).
17. Mather M, Schoeke A. Positive outcomes enhance incidental learning for both younger and older adults. *Frontiers in neuroscience* **5**, 129 (2011).
18. Rouhani N, Norman KA, Niv Y. Dissociable effects of surprising rewards on learning and memory. *Journal of experimental psychology Learning, memory, and cognition* **44**, 1430-1443 (2018).
19. Fiorillo CD, Tobler PN, Schultz W. Discrete coding of reward probability and uncertainty by dopamine neurons. *Science* **299**, 1898-1902 (2003).
20. Davidow JY, Foerde K, Galvan A, Shohamy D. An Upside to Reward Sensitivity: The Hippocampus Supports Enhanced Reinforcement Learning in Adolescence. *Neuron* **92**, 93-99 (2016).
21. Wimmer GE, Braun EK, Daw ND, Shohamy D. Episodic memory encoding interferes with reward learning and decreases striatal prediction errors. *The Journal of neuroscience : the official journal of the Society for Neuroscience* **34**, 14901-14912 (2014).
22. Schultz W, Dayan P, Montague PR. A neural substrate of prediction and reward. *Science* **275**, 1593-1599 (1997).
23. Adcock RA, Thangavel A, Whitfield-Gabrieli S, Knutson B, Gabrieli JD. Reward-motivated learning: mesolimbic activation precedes memory formation. *Neuron* **50**, 507-517 (2006).
24. Shohamy D, Adcock RA. Dopamine and adaptive memory. *Trends in cognitive sciences* **14**, 464-472 (2010).
25. Murty VP, Shermohammed M, Smith DV, Carter RM, Huettel SA, Adcock RA. Resting state networks distinguish human ventral tegmental area from substantia nigra. *Neuroimage* **100**, 580-589 (2014).

26. Aberg KC, Doell KC, Schwartz S. Hemispheric Asymmetries in Striatal Reward Responses Relate to Approach-Avoidance Learning and Encoding of Positive-Negative Prediction Errors in Dopaminergic Midbrain Regions. *The Journal of neuroscience : the official journal of the Society for Neuroscience* **35**, 14491-14500 (2015).
27. Vul E, Harris C, Winkielman P, Pashler H. Puzzlingly High Correlations in fMRI Studies of Emotion, Personality, and Social Cognition. *Perspectives on psychological science : a journal of the Association for Psychological Science* **4**, 274-290 (2009).
28. Nichols T, Brett M, Andersson J, Wager T, Poline JB. Valid conjunction inference with the minimum statistic. *Neuroimage* **25**, 653-660 (2005).
29. Geisler S, Derst C, Veh RW, Zahm DS. Glutamatergic afferents of the ventral tegmental area in the rat. *Journal of Neuroscience* **27**, 5730-5743 (2007).
30. Geisler S, Wise RA. Functional Implications of Glutamatergic Projections to the Ventral Tegmental Area. *Reviews in the neurosciences* **19**, 227-244 (2008).
31. Beier KT, *et al.* Circuit Architecture of VTA Dopamine Neurons Revealed by Systematic Input-Output Mapping. *Cell* **162**, 622-634 (2015).
32. Morales M, Margolis EB. Ventral tegmental area: cellular heterogeneity, connectivity and behaviour. *Nature Reviews Neuroscience* **18**, 73-85 (2017).
33. Canavier CC, Landry RS. An increase in AMPA and a decrease in SK conductance increase burst firing by different mechanisms in a model of a dopamine neuron in vivo. *J Neurophysiol* **96**, 2549-2563 (2006).
34. Floresco SB, West AR, Ash B, Moore H, Grace AA. Afferent modulation of dopamine neuron firing differentially regulates tonic and phasic dopamine transmission. *Nature neuroscience* **6**, 968-973 (2003).
35. Overton PG, Clark D. Burst firing in midbrain dopaminergic neurons. *Brain Res Rev* **25**, 312-334 (1997).
36. Chen M, *et al.* Morphine disinhibits glutamatergic input to VTA dopamine neurons and promotes dopamine neuron excitation. *eLife* **4**, (2015).
37. Grace AA, Bunney BS. The Control of Firing Pattern in Nigral Dopamine Neurons - Burst Firing. *Journal of Neuroscience* **4**, 2877-2890 (1984).

38. Grace AA. Dopamine system dysregulation by the ventral subiculum as the common pathophysiological basis for schizophrenia psychosis, psychostimulant abuse, and stress. *Neurotoxicity research* **18**, 367-376 (2010).
39. Lodge DJ, Grace AA. The hippocampus modulates dopamine neuron responsivity by regulating the intensity of phasic neuron activation. *Neuropsychopharmacology : official publication of the American College of Neuropsychopharmacology* **31**, 1356-1361 (2006).
40. Ikemoto S. Dopamine reward circuitry: Two projection systems from the ventral midbrain to the nucleus accumbens-olfactory tubercle complex. *Brain Res Rev* **56**, 27-78 (2007).
41. D'Ardenne K, McClure S, Nystrom L, Cohen J. BOLD Responses Reflecting Dopaminergic Signals in the Human Ventral Tegmental Area. *Science* **319**, 1264-1267 (2008).
42. Chase HW, Kumar P, Eickhoff SB, Dombrovski AY. Reinforcement learning models and their neural correlates: An activation likelihood estimation meta-analysis. *Cogn Affect Behav Ne* **15**, 435-459 (2015).
43. Diederer KM, Ziauddeen H, Vestergaard MD, Spencer T, Schultz W, Fletcher PC. Dopamine Modulates Adaptive Prediction Error Coding in the Human Midbrain and Striatum. *The Journal of neuroscience : the official journal of the Society for Neuroscience* **37**, 1708-1720 (2017).
44. Kahn I, Shohamy D. Intrinsic connectivity between the hippocampus, nucleus accumbens, and ventral tegmental area in humans. *Hippocampus* **23**, 187-192 (2013).
45. Goto Y, Grace AA. Limbic and cortical information processing in the nucleus accumbens. *Trends in neurosciences* **31**, 552-558 (2008).
46. Floresco SB. The Nucleus Accumbens: An Interface Between Cognition, Emotion, and Action. *Annu Rev Psychol* **66**, 25-52 (2015).
47. FitzGerald THB, Schwartenbeck P, Dolan RJ. Reward-Related Activity in Ventral Striatum Is Action Contingent and Modulated by Behavioral Relevance. *Journal of Neuroscience* **34**, 1271-1279 (2014).
48. Li JA, Daw ND. Signals in Human Striatum Are Appropriate for Policy Update Rather than Value Prediction. *Journal of Neuroscience* **31**, 5504-5511 (2011).
49. Barr DJ, Levy R, Scheepers C, Tily HJ. Random effects structure for confirmatory hypothesis testing: Keep it maximal. *Journal of memory and language* **68**, (2013).

50. Judd CM, Westfall J, Kenny DA. Treating Stimuli as a Random Factor in Social Psychology: A New and Comprehensive Solution to a Pervasive but Largely Ignored Problem. *J Pers Soc Psychol* **103**, 54-69 (2012).
51. Rights JD, Sterba SK. Quantifying explained variance in multilevel models: An integrative framework for defining R-squared measures. *Psychological methods* **24**, 309-338 (2019).

Reviewers' comments:

Reviewer #1 (Remarks to the Author):

The reviewers did a commendable job addressing my concerns from the previous round of revisions. The follow-up analyses they included greatly strengthened the interpretation of the findings. I have a few remaining concerns that are focused on the interpretation of their data given the follow-up analyses that they conducted.

1. The authors state that there is a limitation that they didn't test delayed memory, which suggests that consolidation-related mechanisms aren't driving their results. Rather, they suggest that their effects are influencing encoding. However, this description seems a bit under-specified in the discussion. What specific types of mechanisms do they think that "dopamine" is facilitating at encoding. In rodent data, there is evidence of the stabilization of place fields and lowering of excitability thresholds of hippocampus neurons, while in the human literature there is evidence that dopamine enhances goal-relevant attention and enhances sensitivity to surprise. Could the authors include a few more sentences about what type of encoding processes they believe are driving these effects, through the lens of dopamine signaling.

2. In a follow-up analyses, the authors show that their hippocampal findings across analyses are non-overlapping. However, their interpretation of their data seems more consistent with results that would show overlapping results. How do the authors believe their hippocampal results are related given that they are non-overlapping clusters?

Reviewer #2 (Remarks to the Author):

Authors sufficiently and clearly addressed my concerns from the first round of review. I have only one point to add about sufficiently motivating the hypotheses.

It is unclear what motivated hypotheses for linear- and non-linear effects in NAcc and VTA respectively. I am not sure whether authors meant to be ambivalent about these, but the impression I get from reading Introduction is that they hypothesized linear effects of immediate reward in NAcc and non-linear effects of average reward in VTA. However, I am having a hard time pinpointing what backs up different hypotheses for these ROIs. For example, the hypothesis on the non-linear effects of average reward in VTA (lines 67-72; citations 33-35) is based on studies with patients with Parkinson's Disease patient, in which patients were tested on- and off- dopaminergic meds. DA degeneration in PD is more pronounced in Striatum (which includes NAcc) than VTA, however, studies suggest degeneration is indicated in VTA as well. That being said, we don't know whether the non-linear effects of DA in the patient studies were driven by DA modulation in VTA, NAcc, or the combination of both. Therefore, the cited studies does not provide grounds to hypothesize specifically non-linear effects in VTA as opposed to the linear effects in NAcc. It would make the Introduction much stronger if it is clearly stated what

necessitated those hypotheses. If authors are not positing strong hypotheses and intend to hold an ambivalent stance, Introduction needs to be re-framed as such.

A minor point: I noticed that the terminology of NAcc is mostly consistent now, but still spotted that it was referred to as vSTR in some places (e.g., figure 3).

Reviewers' comments:

Reviewer #1 (Remarks to the Author):

The reviewers did a commendable job addressing my concerns from the previous round of revisions. The follow-up analyses they included greatly strengthened the interpretation of the findings. I have a few remaining concerns that are focused on the interpretation of their data given the follow-up analyses that they conducted.

1. The authors state that there is a limitation that they didn't test delayed memory, which suggests that consolidation-related mechanisms aren't driving their results. Rather, they suggest that their effects are influencing encoding. However, this description seems a bit under-specified in the discussion. What specific types of mechanisms do they think that "dopamine" is facilitating at encoding. In rodent data, there is evidence of the stabilization of place fields and lowering of excitability thresholds of hippocampus neurons, while in the human literature there is evidence that dopamine enhances goal-relevant attention and enhances sensitivity to surprise. Could the authors include a few more sentences about what type of encoding processes they believe are driving these effects, through the lens of dopamine signaling.

We thank you for this additional suggestion. We agree that this topic could be a bit more elaborated upon. We now explicitly refer to two mechanisms that could explain the impact of dopamine on memory encoding. The first mechanism, as you already mentioned, is a dopamine-related lowering of thresholds needed to induce LTP. The second mechanism is related to the three-factor rule of synaptic plasticity, according to which co-activation of pre- and post-synaptic neurons sets an eligibility trace that allows a change in synaptic strength only in the presence of a third modulatory factor, which could be dopamine. Both of these mechanisms contribute to rapid changes in synaptic plasticity independent of consolidation processes.

We added this information to the Discussion (lines: 724-737):

What mechanism(s) could explain reward-influences on memory encoding? Some experimental evidence suggests that dopamine lowers thresholds for LTP induction ¹. Another plausible option relates to the notion of a three-factor rule of synaptic plasticity ². In brief, this theory posits that co-activation of pre- and post-synaptic neurons sets an eligibility trace that allows synaptic change, but only in the presence of a third modulating factor. This modulatory factor may be dopamine, but could also be any other neuromodulator known to impact learning, such as acetylcholine, norepinephrine, or serotonin. This rule explains rapid behavioral change without the need for consolidation, and is supported by recent experimental evidence obtained in the striatum, in the prefrontal and visual cortices, and in the hippocampus (see Gerstner et al., 2018 for a recent review). Other possible mechanisms have been mentioned by Lisman et al. (2011), for example, dopamine may turn on and off different pathways, regulate network oscillations, attenuate inhibition, and/or affect working memory capacity and attentional control (see Box 5 of Lisman et al., 2011). Together, these observations converge to support a role of dopamine activity in mediating enhanced memory encoding for rewarded information.

2. In a follow-up analyses, the authors show that their hippocampal findings across analyses are non-overlapping. However, their interpretation of their data seems more consistent with results that would show overlapping results. How do the authors

believe their hippocampal results are related given that they are non-overlapping clusters?

Thank you for this highly relevant comment. To address this issue, we added a new paragraph to the Discussion which draws upon research showing that tonic and phasic dopamine activity may engage different types of dopamine receptors, which in turn may be distributed heterogeneously across the hippocampus (see lines 547-563):

This interpretation may suggest that the hippocampal loci for the effects of feedback value and immediate reward on memory encoding should overlap, yet a conjunction analysis revealed no significant voxel overlap between the different contrasts. Notably, seen through the lens of phasic and tonic dopamine activity, such a result corroborates suggestions emerging from previous literature. For example, Shohamy and Adcock (2010) proposed that tonic dopamine is more likely to act on extrasynaptic D5 receptors in the hippocampus while phasic dopamine responses are restricted to engage other dopamine receptors within a synapse. Edelmann and Lessman (2018) suggested that tonic dopamine firing only activates high-affinity D2 receptors, while phasic dopamine firing may additionally and briefly activate low-affinity D1 and D5 receptors. Moreover, the flow of information from the CA3 to the CA1 region of the hippocampus (i.e. via Schaffer collaterals) was found to depend on D4 receptors and the current mode of dopamine activity. Specifically, optogenetic stimulation of the midbrain that simulated tonic/phasic mode of dopamine release caused inhibition/facilitation of postsynaptic potentials following Schaffer collateral stimulation³. Given that the densities of different types of dopamine receptors are distributed heterogeneously throughout the hippocampus^{4,5}, these results imply that phasic and tonic dopamine activity may recruit different parts of the hippocampus via the engagement of distinct types of dopamine receptors.

Reviewer #2 (Remarks to the Author):

Authors sufficiently and clearly addressed my concerns from the first round of review. I have only one point to add about sufficiently motivating the hypotheses.

It is unclear what motivated hypotheses for linear- and non-linear effects in NAcc and VTA respectively. I am not sure whether authors meant to be ambivalent about these, but the impression I get from reading Introduction is that they hypothesized linear effects of immediate reward in NAcc and non-linear effects of average reward in VTA. However, I am having a hard time pinpointing what backs up different hypotheses for these ROIs. For example, the hypothesis on the non-linear effects of average reward in VTA (lines 67-72; citations 33-35) is based on studies with patients with Parkinson's Disease patient, in which patients were tested on- and off- dopaminergic meds. DA degeneration in PD is more pronounced in Striatum (which includes NAcc) than VTA, however, studies suggest degeneration is indicated in VTA as well. That being said, we don't know whether the non-linear effects of DA in the patient studies were driven by DA modulation in VTA, NAcc, or the combination of both.

Therefore, the cited studies does not provide grounds to hypothesize specifically non-linear effects in VTA as opposed to the linear effects in NAcc. It would make the Introduction much stronger if it is clearly stated what necessitated those hypotheses. If authors are not positing strong hypotheses and intend to hold an ambivalent stance, Introduction needs to be re-framed as such.

We thank you for raising this important point. We did not have any specific predictions regarding whether linear/non-linear effects of immediate/average reward would be represented in the VTA and/or the NAcc; this is why we included both these regions in the same “reward” mask. We apologize if this was not made sufficiently clear in the Introduction.

To acknowledge your concern, we have now updated the Introduction with the following clarifications:

Lines 72-75:

Yet, it is currently unclear whether these effects of dopamine on memory formation relate to changes in the functionality of the VTA and/or in other brain regions involved in memory formation and motivation, such as the HC and the nucleus accumbens (NAcc).

Lines 81-88:

We predicted that immediate reward magnitude and average reward levels during encoding should engage brain regions involved in reward processing (i.e. the VTA and its downstream target the nucleus accumbens; NAcc) and memory formation (i.e. the hippocampus and the parahippocampal gyrus; PHG), and should account for variance in subsequently tested memory performance. However, because the brain loci enabling a non-linear effect of dopamine on episodic memory formation are unclear, we made no specific predictions regarding a differential neuronal representation of linear versus non-linear effects of immediate and average reward.

Line 90:

We changed the phrasing “Critically, and as predicted ...” to just “Critically, ...”.

A minor point: I noticed that the terminology of NAcc is mostly consistent now, but still spotted that it was referred to as vSTR in some places (e.g., figure 3).

Thank you for spotting this error. We have updated Figure 3. We also went through the manuscript again to ensure that the VStr is only mentioned when referring to terminology used by cited research.

References:

1. Li S, Cullen WK, Anwyl R, Rowan MJ. Dopamine-dependent facilitation of LTP induction in hippocampal CA1 by exposure to spatial novelty. *Nature neuroscience* **6**, 526-531 (2003).
2. Gerstner W, Lehmann M, Liakoni V, Corneil D, Brea J. Eligibility Traces and Plasticity on Behavioral Time Scales: Experimental Support of NeoHebbian Three-Factor Learning Rules. *Frontiers in neural circuits* **12**, 53 (2018).
3. Rosen ZB, Cheung S, Siegelbaum SA. Midbrain dopamine neurons bidirectionally regulate CA3-CA1 synaptic drive. *Nature neuroscience* **18**, 1763-1771 (2015).
4. Edelmann E, Lessmann V. Dopaminergic innervation and modulation of hippocampal networks. *Cell and tissue research* **373**, 711-727 (2018).

5. Shohamy D, Adcock RA. Dopamine and adaptive memory. *Trends in cognitive sciences* **14**, 464-472 (2010).

Reviewer #1 (Remarks to the Author):

The authors addressed all of my remaining concerns that I raised in the last round of revisions.

Reviewer #2 (Remarks to the Author):

All my concerns are sufficiently addressed by authors. It would be exciting to see this addition to literature.

No further issues were raised by the reviewers.